# Biased placement of Mitochondria fission facilitates asymmetric inheritance of protein aggregates during yeast cell division

Gordon Sun[1,2], Christine Hwang[3], Tony Jung[3], Jian Liu[1,4]*, Rong Li[1,4,5,6]*

1 Center for Cell Dynamics and Department of Cell Biology, Johns Hopkins University School of Medicine, Baltimore, Maryland, United States of America, 2 Biomedical Engineering Graduate Program, Johns Hopkins University School of Medicine, Baltimore, Maryland, United States of America, 3 Johns Hopkins University, Baltimore, Maryland, United States of America, 4 Biochemistry, Cellular and Molecular Biology Graduate Program, Johns Hopkins University School of Medicine, Baltimore, Maryland, United States of America, 5 Mechanobiology Institute, National University of Singapore, Singapore, Singapore, 6 Department of Biological Sciences, National University of Singapore, Singapore, Singapore

* jliu187@jhmi.edu (JL); rong@jhu.edu (RL)

**Data Availability Statement:** All data are in the manuscript and/or supporting information files. Simulation code is stored on Github (https://github.

## Abstract

Mitochondria are essential and dynamic eukaryotic organelles that must be inherited during cell division. In yeast, mitochondria are inherited asymmetrically based on quality, which is thought to be vital for maintaining a rejuvenated cell population; however, the mechanisms underlying mitochondrial remodeling and segregation during this process are not understood. We used high spatiotemporal imaging to quantify the key aspects of mitochondrial dynamics, including motility, fission, and fusion characteristics, upon aggregation of misfolded proteins in the mitochondrial matrix. Using these measured parameters, we developed an agent-based stochastic model of dynamics of mitochondrial inheritance. Our model predicts that biased mitochondrial fission near the protein aggregates facilitates the clustering of protein aggregates in the mitochondrial matrix, and this process underlies asymmetric mitochondria inheritance. These predictions are supported by live-cell imaging experiments where mitochondrial fission was perturbed. Our findings therefore uncover an unexpected role of mitochondrial dynamics in asymmetric mitochondrial inheritance.

## Author summary

In this study, we sought to unravel the complex process of how cells segregate damaged parts of mitochondria, ensuring the healthier mitochondria to be passed on to progeny cells. Using budding yeast as a model organism, we characterized how the presence of misfolded protein aggregates in mitochondrial matrix alters mitochondria remodeling dynamics, leading to an asymmetric retention of these protein aggregates in the aging mother cell. Through a combination of stochastic agent-based modeling and high spatiotemporal microscopy imaging, we found that aggregated misfolded proteins inside mitochondria cause mitochondria to fission unevenly. This skewed division, or "biased fission", promotes the clustering of aggregates, which facilitates their retention by the

com/RongLiLab/Sun-et-al._mito_inheritance_simulation). Data used for generating figures. plasmid, and strain sequences are stored on figshare at (doi.org/10.6084/m9.figshare.24550477). Additional data related to this paper can be requested from the JHU Department of Biochemistry, Cellular and Molecular Biology at bcmb@jhmi.edu.

**Funding:** This was work was supported by a grant from ReStem Biotech grant to R.L., J.L. was supported by start-up funds from the Johns Hopkins University School of Medicine, Johns Hopkins Catalyst award, the National Science Foundation (2105837 and 2148534), and National Institutes of Health (1RO1 GM148459-01). G.S. was supported by an NDSEG Fellowship. Research reported in this publication was supported by the Office of the Director and the National Institute of General Medical Sciences of the National Institutes of Health under award number S10OD023548. Computing work was carried out at the Advanced Research Computing at Hopkins (ARCH) core facility (rockfish.jhu.edu), which is supported by the National Science Foundation (NSF) grant number OAC1920103. The funders had no role in study design, data collection and analysis, decision to publish, or preparation of the manuscript.

**Competing interests:** None.

mother cells. By genetically perturbing key components of fission machinery, we show that impaired mitochondrial fission reduces the rate of aggregate clustering and elevates their propagation into the daughter cell.

## Introduction

Asymmetric inheritance is a non-Mendelian strategy where a cell segregates its cellular components unequally during cell division, resulting in non-identical progeny cells [1]. In multicellular eukaryotes, such as humans, asymmetric inheritance plays a vital role in many aspects of biology, such as stem cell differentiation [2,3], cellular ageing [4], and immunological memory formation [5,6]. Similarly, unicellular organisms such as the budding yeast *Saccharomyces cerevisiae* also exhibit asymmetric inheritance of cellular components, including but not limited to protein aggregates [7–11], mitochondria [12–14], extrachromosomal rDNA circles [15], and vacuoles [16,17]. As a result, one of the progeny cells, the bud, from each division are born age-free, irrespective of the age of their mother cell, thus maintaining an age-rejuvenated population of cells with high proliferative potential [1,4]. Without asymmetric division, the long-range functional activity of tissue and the continued survival of a population of cells theoretically can be compromised [18].

Asymmetric inheritance of mitochondria is one manifestation of asymmetric cell division helping maintain cellular fitness and may serve as a filter against mitochondrial dysfunction. This process may be particularly crucial in the context of cellular disease, where the accumulation of aggregated proteins is a common feature observed in many neurodegenerative diseases [19–21]. In budding yeast, asymmetric inheritance of mitochondria involves the preferential passage of mitochondria with higher redox potential to the bud, while low redox mitochondria are retained by the mother cell [22]. Impairment of mitochondrial quality may be due to accumulated proteome damage, which may be reflected in the accumulation of misfolded-protein aggregates inside or outside of mitochondria [23,24]. Previous studies in yeast showed that mitochondria bearing protein aggregates are preferentially retained in the mother cell, whereas mitochondria inherited by the daughter cell are damage-free [8,22,24,25]. There is also emerging evidence that a similar pattern of asymmetric mitochondria inheritance exists in mammalian cells [2,12].

To achieve this selective segregation, a remodeling process involving sorting and filtering of mitochondria with distinct properties is essential to prevent random inheritance during cell division [26–28] In yeast and many metazoan cell types, mitochondria organize themselves as a well-distributed network of tubular structures that constantly undergo fission, fusion, and diffusive or directed motion [29–31]. Previous studies suggest that multiple mechanisms, such as remodeling of the mitochondrial network [29,32–36], selective anchoring [37–40], and retrograde actin cable flow and anterograde mitochondria motion during passage, may be involved in the selective inheritance of mitochondria [41,42]. However, understanding the precise role of mitochondrial remodeling in asymmetric inheritance is challenging, due to the intricate interplay between individual facets of mitochondria dynamics. Whereas mutants lacking core elements of either mitochondria fission, fusion, or trafficking machinery exhibit significant abnormalities in mitochondrial inheritance [43–46], perturbations in the proteins involved can have confounding effects such as altering mitochondria morphology and distribution [47,48], ER-contact sites [49–51], and a wide spectrum of cellular processes [52–54]. This complexity necessitates the combination of modeling and experiments to identify quantitative determinants of asymmetric mitochondria inheritance.

In this study, we integrated live-cell experiments and modeling to determine how mitochondrial remodeling underlies asymmetric mitochondria inheritance. Using budding yeast cells expressing mitochondria-targeted misfolded proteins as a model system, we first quantified the parameters previously suggested to critically impact mitochondrial inheritance, such as mitochondria fission/fusion kinetics and motility. Using these measured parameters as constraints, we then developed an agent-based stochastic model of mitochondria inheritance to predict the impact of mitochondrial dynamics on mitochondria inheritance. Our model predicted that biasing mitochondrial fission close to the protein aggregates plays a crucial role in aggregate clustering, which improves retention of the dysfunctional mitochondria in the mother cell. This prediction was supported by our *in vivo* experiments with mutants lacking mitochondrial fission machinery.

## Results

### Protein aggregates alter mitochondrial remodeling dynamics during asymmetric inheritance

To quantitatively characterize how mitochondria remodeling correlates with asymmetric mitochondria inheritance, we used a previously established experimental model using budding yeast cells. In this model, a constitutively expressed, structurally unstable protein (fly luciferase R188Q, or FlucSM) [55] fused to mCherry is trafficked into the mitochondria matrix via a mitochondrial matrix targeting sequence (MTS) [56]. Once in the matrix, MTS-FlucSM-mCherry (referred to herein as mitoFluc) forms distinct deposits of unfolded mitochondrial proteins (DUMP) [24]. We grouped mitochondria into three classes: (1) wild-type (WT), which are in cells of the same parental genetic background but did not express mitoFluc, and in mitoFluc-expressing cells (2) DUMP⁻ for mitochondria that did not contain DUMP and (3) DUMP⁺ for mitochondria that contained DUMP (Fig 1A). In all strains, Tom70-GFP was used as the marker for mitochondria.

As expected, over multiple budding cycles, the mother cell retained mitochondria containing DUMP (or equivalently, DUMP⁺ mitochondria), preferentially passing the mitochondria without DUMP (DUMP⁻) to the daughter cell (Fig 1B). Given that ~55% of mitochondria contain DUMP (S1A Fig), the observed 15% inheritance rate of DUMP(s) by the daughter cell confirmed that mitochondrial inheritance is not a random process (Fig 1C).

Importantly, using high-spatiotemporal-resolution imaging we noticed that patterns of mitochondria remodeling (fusion and fission) correlated with the presence or absence of DUMP. DUMP⁺ mitochondria were more likely to undergo fission than either DUMP⁻ or WT mitochondria (Fig 1D). Although prior studies suggested that membrane potential is important for fusion [57–59], both DUMP⁻ and DUMP⁺ mitochondria exhibited reduced membrane potential compared to WT mitochondria, as indicated by DiOC6 staining (S1B and S1C Fig). Interestingly, DUMP⁺ mitochondria exhibited a fusion rate similar to WT mitochondria, while DUMP⁻ mitochondria were more fusion prone (Fig 1E). These observations suggest that the presence of DUMP has differential effects on fission and fusion independent of membrane potential.

We next quantified how mitochondrial motility correlates with the presence of DUMPs within mitochondria. We first focused on the diffusive motion of mitochondria, which is their primary mode of motility prior to inheritance in the bud [47,60]. Our results showed that DUMP⁻ mitochondria exhibited significantly greater diffusive mobility than either WT or DUMP⁺ mitochondria (Fig 1F). This suggests that while the presence of DUMP does not impact on the mobility of mitochondria relative to WT, it leads to the emergence of two distinct mitochondria populations exhibiting different mobility.

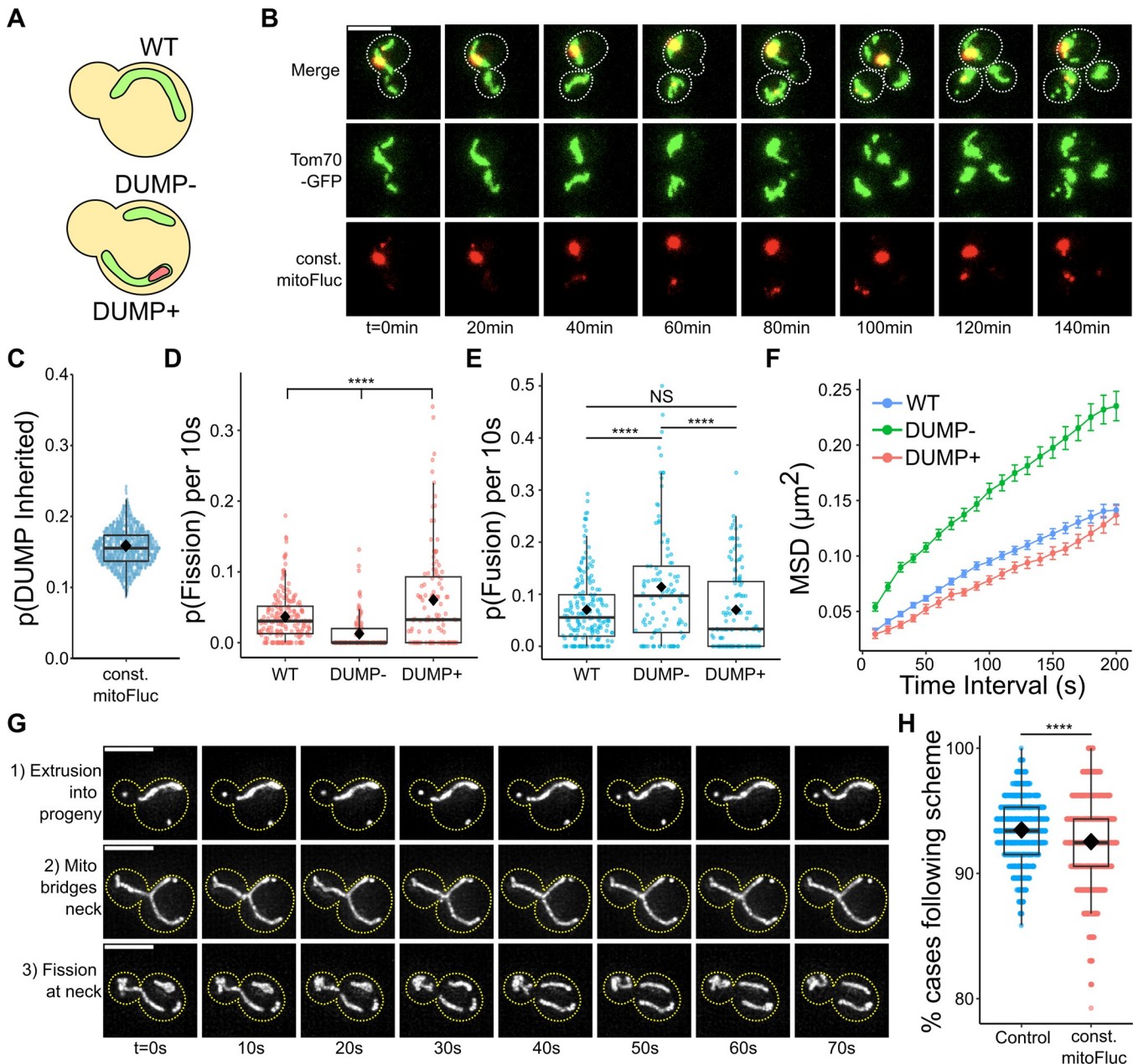

**Fig 1. Mitochondrial aggregates are asymmetrically inherited and alter mitochondrial remodeling probabilities and diffusive properties. Mitochondrial inheritance follows a stereotyped three step process.** (A) Definition of mitochondria subtypes by DUMP status. (B) Time series of mitoFluc retention by mother cell over 180 min. Mitochondria are labeled using Tom70-GFP. For all cell images unless otherwise noted: scale bar = 5µm in first merge image; white dashed line demarcates cell boundaries traced from DIC images; all images portrayed are maximum intensity projections of 3D stack images. (C) Probability of inheritance of mitoFluc aggregate (DUMP) by bud. Each point is indicative of bootstrapped p(DUMP Inherited) from a population of 219 cells, sampled with batch size = 219, iterations = 1000. Black diamond = mean for this and all subsequent box plots. (D) Probability of mitochondria to undergo fission, fusion (E) per 10s by mitochondria type. NS = Not Significant (p = 0.9995). (D-E) Tukey's HSD multiple comparisons test. ****p<0.0001. n = 134 single cell movies for mitoFluc, n = 208 for control (Tom70-GFP). Each point represents a single cell's population of the associated mitochondria type. (F) MSD curves for mitochondria types; D = 7.809x10$^{-4}$, 1.248x10$^{-3}$, 7.964x10$^{-4}$ µm$^2$/s, for WT, DUMP-, DUMP+ mitochondria respectively. n = 184, 120 single cell movies respectively for WT and mitoFluc strain respectively. N≥2558, 677, 819 MSD trajectories for WT, DUMP-, DUMP+ mitochondria respectively. Mean±SEM (standard error of mean). (G) Mitochondria inheritance process. WT shown (Tom70-GFP). Yellow dashed line demarcates cell boundaries traced from DIC images. (H) Bootstrapped proportion of mitochondrial inheritance events following scheme in (D). Bootstrap iterations = 1000, sample size = n; n = 137, 87 cells for WT and constitutive mitoFluc strain respectively. T-test, ****p<0.0001.

Finally, we monitored the motility pattern as the mitochondrial network extends into the emerging bud. Time-lapse imaging identified a distinct two-step process for mitochondrial inheritance (Fig 1G). First, the largest mitochondria in the progenitor cell extrudes the longest branch of itself into the bud, forming a mitochondria bridge. Second, this bridge undergoes fission at the bud neck, thereby isolating a piece of mitochondria in the bud. Importantly, this process remained consistent regardless of mitoFluc expression, with greater than 90% of mitochondria inheritance events following this observed pattern (Fig 1H).

The above observations remained consistent even when accounting for the varying volumes of mitochondria. Both the rates of fission and fusion (S1D–S1F Fig), and mitochondria diffusive motility (S1G–S1I Fig) were proportionate to their volume. These experiments suggest the presence of DUMP differentially affects mitochondria diffusion and fission/fusion, which could potentially facilitate asymmetric DUMP retention through mitochondrial sorting prior to their extrusion into the bud.

For controls, constitutively expressed MTS-mCherry or MTS-Fluc-mCherry (mitoFluc (WT)) showed a uniform mCherry signal throughout the mitochondrial matrix mirroring the mitochondria network structure (S2A and S2B Fig). Importantly, these constructs did not alter the mitochondrial membrane potential or the rate of fission or fission (S2C, S2D, S3A and S3B Figs). However, when we induced expression of the mutant form, MTS-FlucSM-mCherry (mitoFluc) using an β-Estradiol inducible promoter, there was a noticeable deviation from the baseline fission/fusion rates 30 minutes post-induction, coinciding with DUMP formation (S3C and S3D Fig). These findings combined with our previous data suggest that the changes in mitochondrial dynamics were directly linked to DUMP formation.

## Developing agent-based stochastic model of mitochondrial inheritance

We next investigated how DUMP-mediated changes in mitochondria fission/fusion kinetics and diffusive motion may help ensure the asymmetric retention of DUMP$^+$ mitochondria in the mother cells. We exploited mathematical modeling in combination with experiments to dissect how mitochondrial remodeling dynamics underlies asymmetric inheritance. Accordingly, we constructed an agent-based stochastic model of mitochondria inheritance, wherein the model parameters are quantitatively constrained by our experimental measurements, including but not limited to mitochondria fission/fusion rates and diffusion constants (Table 1).

To capture the essence of mitochondria remodeling dynamics, we modeled the mitochondria network as a collection of 2D non-deformable, indivisible particles, each representing a unit of mitochondrion. Within each simulation time step, these units may undergo one of three dynamical processes. First, particles can fuse with each other to form networked structures at a fusion rate, $K_{fus}$. Second, particles can detach from a network structure at a fission rate, $K_{fis}$. Last, particles can diffuse at the diffusion constant, $D$. Networked structures formed by these particles likewise undergo these processes. We simulated these processes stochastically by the kinetic Monte Carlo algorithm [61].

To ensure the relevance of our model to the *in vivo* process, we calibrated the size of each particle to the experimentally determined minimal size of a mitochondrion (Table 1). This size determination was accomplished utilizing a Tom70-GFP, Δfzo1 strain. Fzo1 is a mitochondria outer membrane protein required for fusion of mitochondria [43,62]. By deleting the *FZO1* gene, mitochondria fragment and remain as smaller individual units. Utilizing this strain allowed us to investigate the distribution of fragmented mitochondria volumes (S4A and S4B Fig). The radius of the mitochondrion unit, $r_{particle}$, was determined from the average volume of fragmented mitochondria ($\bar{V}_{\Delta fzo1}^{mito}$) by assuming it as a sphere in 3D. There are 54

**Table 1. Experimentally measured and calculated parameters.**

| Parameter | Symbol | Experimentally measured value | Value used for simulation | Units | Figure |
|---|---|---|---|---|---|
| Mean volume of unit mitochondria (Δfzo1) | $\bar{V}^{mito}_{\Delta fzo1}$ | 0.12±0.14 | 0.12 | μm³ | S4B |
| Mean volume of progenitor cell | $\bar{V}^{cell}_{mom}$ | 54.04±14.83 | N/A | μm³ | S4C |
| Mean total volume of mitochondria | $\bar{V}^{mito}_{total}$ | 6.56±2.01 | 6.56 | μm³ | S4C |
| Mean mitochondria volume in progeny | $\bar{V}^{mito}_{bud}$ | 1.83±1.57 | 1.80 ($N_{inherit} \times \bar{V}^{mito}_{\Delta fzo1}$) | μm³ | S4D |
| Average % mitochondrial volume occupied by DUMP | $P_{occupancy}$ | 12.2±6.1 | 12.2 | % | S4E |
| Computed progenitor cell radius | $r_{mom}$ | 2.35 ($\sqrt[3]{3\bar{V}^{cell}_{mom}/4\pi3}$) | NA | μm | |
| Computed progenitor cell surface area | $SA_{mom}$ | 69.12 ($4\pi r^2_{mom}$) | 69.12 | μm² | |
| Computed side length of 2D square from progenitor cell surface area | L | 8.31 ($\sqrt{SA_{mom}}$) | 8.31 | μm | |
| Volume of unit mitochondrion in simulation | $V^{sim}_{particle}$ | NA | 0.12 ($\bar{V}^{mito}_{\Delta fzo1}$) | μm³ | |
| Radius of unit mitochondria particle | $r_{particle}$ | NA | 0.31 ($\sqrt[3]{3V^{sim}_{particle}/4\pi3}$) | μm | |
| Total number of unit mito in simulation | $N_{total}$ | NA | 54 ($\sim \bar{V}^{mito}_{total}/\bar{V}^{mito}_{\Delta fzo1}$) | count | |
| Number of mito particles inherited | $N_{inherit}$ | NA | 15 ($\sim \bar{V}^{mito}_{bud}/\bar{V}^{mito}_{\Delta fzo1}$) | count | |
| Number of DUMP particles to seed | $N_{DUMP}$ | NA | 6 (~P×N) | count | |
| Total mitochondria volume vs. Progenitor cell volume | $V^{mito}_{total}(V^{cell}_{mom})$ | Slope: **** 6.27×10⁻²±0.65×10⁻² Intercept: **** 3.07±0.36 RSE: 1.789 | NA | unitless μm³ unitless | S4C |
| Progeny mitochondria volume vs. progenitor cell volume | $V^{mito}_{bud}(V^{cell}_{mom})$ | Slope: **** 2.23×10⁻²±0.56×10⁻² Intercept: * 0.63±0.31 RSE: 1.537 | NA | unitless μm³ unitless | S4D |
| Rate of fission in WT mitochondria normalized by mito volume † | $K^{WT}_{fis}(v)$ | Slope: **** 1.28×10⁻³±0.13×10⁻³ Intercept: -1.67×10⁻³±0.87×10⁻³ RSE: 2.56×10⁻³ | Slope: 1.28×10⁻³ Intercept: -1.67×10⁻³ | μm⁻³s⁻¹ s⁻¹ μm⁻³s⁻¹ | S1D |
| Rate of fusion in WT mitochondria normalized by mito volume † | $K^{WT}_{fus}(v)$ | Slope: **** -3.72×10⁻⁴±0.59×10⁻⁴ Intercept: **** 8.04×10⁻³±0.69×10⁻³ RSE: 2.02×10⁻³ | Slope: -3.72×10⁻⁴ Intercept: 8.04×10⁻³ | μm⁻³s⁻¹ s⁻¹ μm⁻³s⁻¹ | S1D |
| Rate of fission in DUMP- mitochondria normalized by mito volume † | $K^{DUMP-}_{fis}(v)$ | Slope: **** 1.57×10⁻³±0.01×10⁻³ Intercept: **** -5.48×10⁻⁴±1.48×10⁻⁴ RSE: 2.31×10⁻³ | Slope: 1.57×10⁻³ Intercept: -5.48×10⁻⁴ | μm⁻³s⁻¹ s⁻¹ μm⁻³s⁻¹ | S1E |
| Rate of fusion in DUMP- mitochondria normalized by mito volume † | $K^{DUMP-}_{fus}(v)$ | Slope: **** -7.93×10⁻⁴±1.24×10⁻⁴ Intercept: **** 1.19×10⁻²±0.03×10⁻² RSE: 4.61×10⁻³ | Slope: -7.93×10⁻⁴ Intercept: 1.19×10⁻² | μm⁻³s⁻¹ s⁻¹ μm⁻³s⁻¹ | S1E |
| Rate of fission in DUMP+ mitochondria normalized by mito volume † | $K^{DUMP+}_{fis}(v)$ | Slope: **** 1.19×10⁻³±0.12×10⁻³ Intercept: 0.13 -1.33×10⁻³±0.84×10⁻³ RSE: 2.01×10⁻³ | Slope: 1.19×10⁻³ Intercept: -1.33×10⁻³ | μm⁻³s⁻¹ s⁻¹ μm⁻³s⁻¹ | S1F |
| Rate of fusion in DUMP+ mitochondria normalized by mito volume † | $K^{DUMP+}_{fus}(v)$ | Slope: NS, p = 0.413 -1.59×10⁻⁴±1.90×10⁻⁴ Intercept: **** 7.15×10⁻³±1.31×10⁻³ RSE: 3.12×10⁻³ | Slope: -1.59×10⁻⁴ Intercept: 7.15×10⁻³ | μm⁻³s⁻¹ s⁻¹ μm⁻³s⁻¹ | S1F |

*(Continued)*

**Table 1.** (Continued)

| Parameter | Symbol | Experimentally measured value | Value used for simulation | Units | Figure |
|---|---|---|---|---|---|
| Diffusive motion constant of WT mitochondria normalized by mito volume † | $D_{WT}(v)$ | Slope: **** <br> $-3.90 \times 10^{-5} \pm 1.56 \times 10^{-5}$ <br> Intercept: * <br> $9.50 \times 10^{-4} \pm 1.22 \times 10^{-4}$ <br> RSE: $1.93 \times 10^{-4}$ | Slope: **** <br> $-3.90 \times 10^{-5}$ <br> Intercept: * <br> $9.50 \times 10^{-4}$ | $\mu m^{-1} s^{-1}$ <br><br> $\mu m^2 s^{-1}$ <br> $\mu m^2 s^{-1}$ | S1G |
| Diffusive motion constant of DUMP- mitochondria as a function of volume † | $D_{DUMP-}(v)$ | Slope: NS, p = 0.1017 <br> $1.74 \times 10^{-4} \pm 0.82 \times 10^{-4}$ <br> Intercept: ** <br> $1.32 \times 10^{-3} \pm 0.25 \times 10^{-3}$ <br> RSE: $3.44 \times 10^{-4}$ | Slope: <br> $1.74 \times 10^{-4}$ <br> Intercept: <br> $1.32 \times 10^{-3}$ | $\mu m^{-1} s^{-1}$ <br><br> $\mu m^2 s^{-1}$ <br> $\mu m^2 s^{-1}$ | S1H |
| Diffusive motion constant of DUMP+ mitochondria as a function of volume † | $D_{DUMP+}(v)$ | Slope: * <br> $-4.10 \times 10^{-5} \pm 1.34 \times 10^{-5}$ <br> Intercept: *** <br> $9.57 \times 10^{-4} \pm 1.05 \times 10^{-4}$ <br> RSE: $1.93 \times 10^{-4}$ | Slope: <br> $-4.10 \times 10^{-5}$ <br> Intercept: <br> $9.57 \times 10^{-4}$ | $\mu m^{-1} s^{-1}$ <br><br> $\mu m^2 s^{-1}$ <br> $\mu m^2 s^{-1}$ | S1I |
| Simulation timestep size | $dt_{sim}$ | NA | 0.1 | s | |
| Experimental movie timestep size | $dt_{expt}$ | 10 | NA | s | |
| Simulation equilibration time | $T_{equil}$ | NA | 1800 | s | |
| Simulation evaluation period run time | $T_{eval}$ | NA | 5400 | s | |
| Simulation total run time | $T_{tot}^{sim}$ | NA | 7200 | s | |

Table of experimentally measured parameters and the values used for simulations

*p≤0.05

**p≤0.01

***p≤0.001

****p≤0.0001

NS = not significant, h value is equal to zero. NA = not applicable; RSE = residual standard error.

† Since the simulation models mitochondria as 2D particles, to get the volume of a mitochondria network, the number of particles in the network is multiplied by $\bar{V}_{\Delta fzo1}^{mito}$ to get the equivalent volume. This volume is then used for calculating $K_{fis,fus}(v)$ and $D(v)$. All equations are linear models fit using least squares regression.

mitochondrion particles in the model, a number determined by dividing the average total mitochondria volume for an average budding progenitor ($\bar{V}_{total}^{mito}$) cell by the volume of an individual particle ($\bar{V}_{\Delta fzo1}^{mito}$, Figs 2A and S4C). Importantly, $K_{fus}$, $K_{fis}$ and $D$ are measured parameters that scale with the size of mitochondria network (*i.e.*, the number of mitochondrion units in the network), as evidenced in our experimental data (Table 1, S1D–S1I Fig).

Using the above model ingredients, we first generated WT-like mitochondria network structures in the model as the baseline for mitochondrial inheritance. We randomly positioned the 54 mitochondrion units in a simulated domain (a square of area $SA_{mom}$) with periodic boundary conditions, mirroring the geometry of the mother cell (Table 1). We then let them self-organize into network structures by fission, fusion, and diffusion at the rates corresponding to those measured in WT (without the expression of protein aggregates). After stochastic simulation of this self-organization process for 30 minutes, we considered the resulting mitochondrial networks as WT-like (Fig 2A), since the resulting number of mitochondrial networks stabilized (S5A Fig).

Using the resulting mitochondrial networks as the starting point, we proceeded to investigate mitochondrial inheritance. Six mitochondrial units were selected based on the average percentage of mitochondria occupied by DUMP (Table 1, S4E Fig). To emulate the distribution of DUMPs per cell as observed in constitutive mitoFluc cells, these six DUMP-containing units were positioned to form N clusters. Here, N corresponds to the number of DUMPs per

 

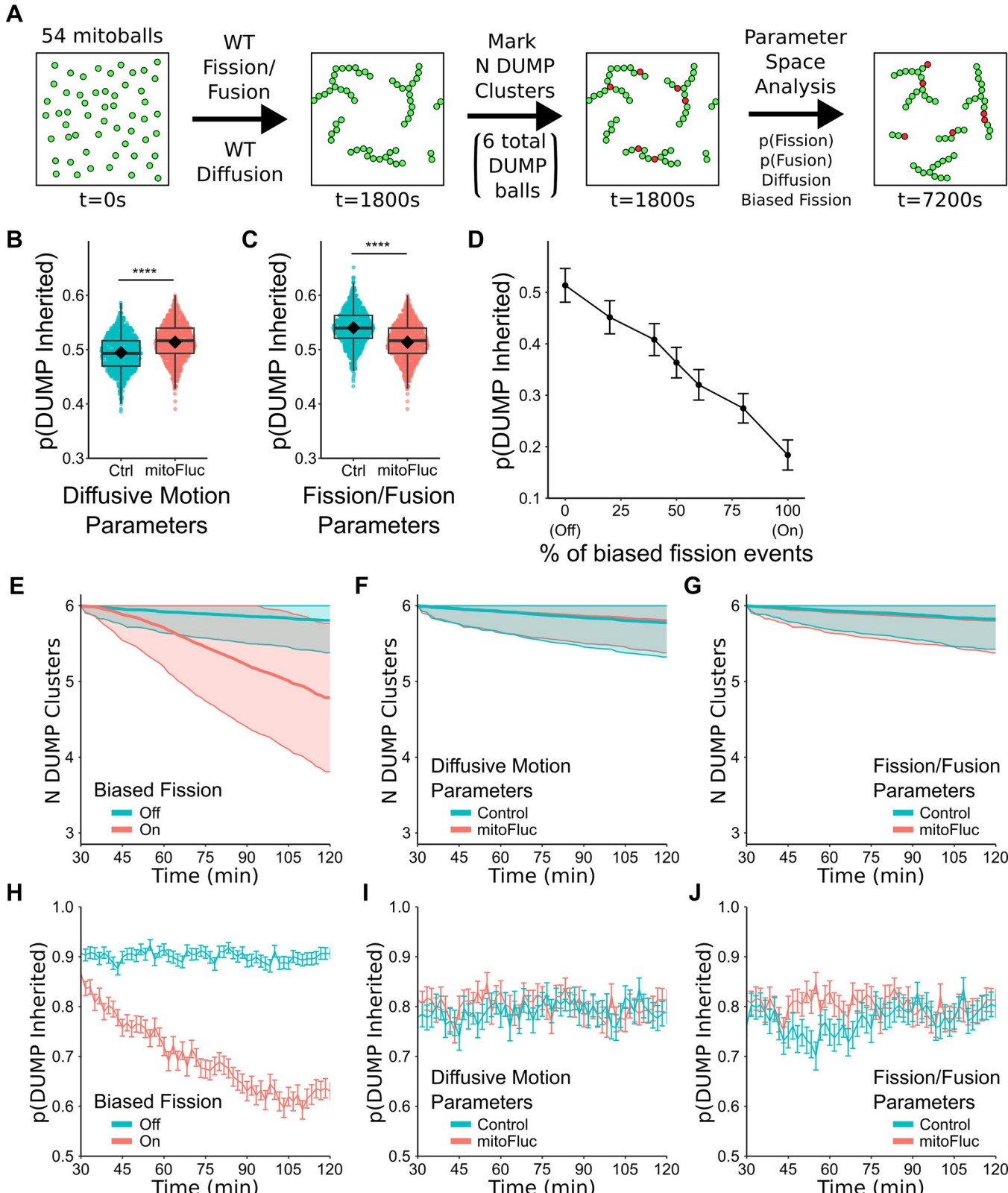

**Fig 2. Agent based modeling of mitochondrial inheritance predicts number of DUMPs is correlated with DUMP inheritance, and biased fission has the largest impact on reducing probability of inheritance.** (A) Model schematic and run procedure. See methods for more details. Not drawn to scale. (B-E) Probability of DUMP inheritance by bud at end of simulation (7200s). Two sample t-test, ****p<0.0001. n = 1000 simulations per boxplot. (B-D) At t = 1800s,

 

N DUMP clusters were seeded, where N follows the observed distribution of number of DUMP(s) in the constitutive mitoFluc strain (S4F Fig) (B) Model results with control ($D_{WT}$) vs. mitoFluc ($D_{DUMP+/-}$) diffusive motion, $K_{fis}^{DUMP+/-}$, $K_{fus}^{DUMP+/-}$ fixed for both control and mitoFluc. (C) Model results with control ($K_{fis}^{WT}$, $K_{fus}^{WT}$) vs. mitoFluc ($K_{fis}^{DUMP+/-}$, $K_{fus}^{DUMP+/-}$) fission/fusion rates, $D_{DUMP+/-}$ fixed for both control and mitoFluc. Biased fission is off for (B, C), with n = 140 simulations. (D) Effect of biased fission on bootstrapped probability of DUMP inheritance. When percentage of fission events that are biased is 100%, all fission events are biased (equivalent to biased fission is on) and vice versa for 0%. Results with $K_{fis}^{DUMP+/-}$, $K_{fus}^{DUMP+/-}$, $D_{DUMP+/-}$ fixed in model. n = 140 simulations for tested percentages 0, 20, 40, 50, 60, 80, 100%. (E-J) Total number of DUMP particles and clusters was fixed to 6, to measure clustering effect if all DUMPs started out as separate entities (E) Number of DUMP clusters in simulation over time with associated (H) bootstrapped probability of DUMP inheritance over time with biased fission on vs. off. (E, H) n = 643, 645 simulations respectively, with fixed $K_{fis}^{DUMP+/-}$, $K_{fus}^{DUMP+/-}$, $D_{DUMP+/-}$. (F) Number of DUMP clusters in simulation over time with associated (I) bootstrapped probability of DUMP inheritance over time with WT ($D_{WT}$) vs. mitoFluc ($D_{DUMP+/-}$) diffusive motion. (F, I) n = 645, 643 simulations respectively, with no biased fission and fixed $K_{fis}^{DUMP+/-}$, $K_{fus}^{DUMP+/-}$. (G) Number of DUMP clusters in simulation over time with associated (J) bootstrapped probability of DUMP inheritance over time with WT ($K_{fis}^{WT}$, $K_{fus}^{WT}$) vs. mitoFluc ($K_{fis}^{DUMP+/-}$, $K_{fus}^{DUMP+/-}$) fission/fusion. (G, J) n = 645,645 simulations respectively, with no biased fission and fixed $D_{DUMP+/-}$. Mean±standard deviation (SD) shown for (D, H-J). For all bootstrap probabilities sampling was done with iterations = 1000, sample size = number of simulations run.

cell. Across all simulations, these N clusters mirrored the distribution of DUMPs typically seen in constitutive mitoFluc cells (S4F Fig). From this point onward, the model's mitochondrial fission/fusion rates and diffusion constant were changed to the measured values in the DUMP expression conditions (Table 2) to simulate the remodeling of mitochondrial networks in response to protein aggregation. Following a simulated period of 90 minutes, which corresponds to the average cell cycle time for yeast under our imaging conditions [63], we assessed mitochondria inheritance.

Based on experimental observations (Fig 1G and 1H), we implemented a two-step process of mitochondria inheritance in our model at the 90-minute timepoint. First, we identified the largest mitochondria network in the simulation at 90-minute timepoint. Next, we identified the longest branch of this mitochondria network and selected $N_{inherit}$ particles from the tip of the branch for inheritance in the bud (S5B Fig). This procedure mimics our experimental

**Table 2. Parameter Space Search.**

| Name (If applicable) | Fission/Fusion Probability | Diffusive Motion | Biased Fission* |
|---|---|---|---|
| Equilibrating Conditions (also WT parameters) | $K_{fis}^{WT}$, $K_{fus}^{WT}$ | $D_{WT}$ | Off (0) |
| mitoFluc parameters | $K_{fis}^{DUMP-}$, $K_{fus}^{DUMP-}$ <br> $K_{fis}^{DUMP+}$, $K_{fus}^{DUMP+}$ | $D_{DUMP-}$ <br> $D_{DUMP+}$ | On (100) |
| | $K_{fis}^{DUMP-}$, $K_{fus}^{DUMP-}$ <br> $K_{fis}^{DUMP+}$, $K_{fus}^{DUMP+}$ | $D_{DUMP-}$ <br> $D_{DUMP+}$ | On** |
| | $K_{fis}^{DUMP-}$, $K_{fus}^{DUMP-}$ <br> $K_{fis}^{DUMP+}$, $K_{fus}^{DUMP+}$ | $D_{DUMP-}$ <br> $D_{DUMP+}$ | Off (0) |
| | $K_{fis}^{DUMP-}$, $K_{fus}^{DUMP-}$ <br> $K_{fis}^{DUMP+}$, $K_{fus}^{DUMP+}$ | $D_{WT}$ | Off (0) |
| | $K_{fis}^{WT}$, $K_{fus}^{WT}$ | $D_{DUMP-}$ <br> $D_{DUMP+}$ | Off (0) |
| | $K_{fis}^{DUMP-}$, $K_{fus}^{DUMP-}$ <br> $K_{fis}^{DUMP+}$, $K_{fus}^{DUMP+}$ | $D_{WT}$ | On** |
| | $K_{fis}^{WT}$, $K_{fus}^{WT}$ | $D_{DUMP-}$ <br> $D_{DUMP+}$ | On** |
| | $K_{fis}^{WT}$, $K_{fus}^{WT}$ | $D_{WT}$ | On** |

List of parameter sets used in simulation.

* The value enclosed by parentheses indicates the percentage of fission events that were set to biased.

** A range of values (0, 20, 40, 50, 60, 80, 100)% was tested.

observations of mitochondrial extrusion into the bud (Fig 1H). Additionally, the value of $N_{inherit}$ in our model was determined by the average mitochondria volume in the bud (S4D Fig), which we measured from experimental images of budding cells (Table 1).

Using this algorithm for inheritance, we next calculated the probability of DUMP$^+$ mitochondria inheritance by the bud. With each set of model parameters, the probability of DUMP inheritance was calculated by averaging over multiple independently and stochastically simulated trajectories.

## Biased fission is predicted to underlie the asymmetric retention of DUMP via DUMP clustering

Stochastic simulations of our model with the above measured parameters showed that the DUMP-mediated changes in mitochondria fission/fusion rates and diffusion were found, surprisingly, insufficient to ensure asymmetric DUMP retention (Fig 2). In fact, adjusting the mitochondria fission/fusion rates (Fig 2B) or diffusive motion (Fig 2C) did little to improve the probability of DUMP inheritance. This indicates that these parameters may not be the primary factors influencing retention of DUMP$^+$ mitochondria. We therefore hypothesize that the localized effects of DUMP, rather than the global changes in mitochondrial remodeling dynamics, may alter the probability of DUMP inheritance. While DUMP alters the rates of mitochondrial fission and fusion, the specific positioning of these fission/fusion events may have an influence on the reorganization of mitochondria [24,64,65]. Indeed, we observed mitochondria undergoing fission close to where DUMPs are localized (Figs 1B and 3A).

Based on this observation, we next utilized the model to examine the potential influence of biased fission on DUMP inheritance outcomes. Under normal circumstances, fission placement was random. When biased fission was present, it would only occur between adjacent DUMP marked particles and non-DUMP marked particles. We then evaluated how the percentage of the biased fission events impacts DUMP inheritance. Our model predicted that increasing the percentage of biased fission events decreased the probability of DUMP inheritance by the bud (Fig 2D). To match the observed probability of DUMP inheritance (Fig 1C), our model predicted that the percentage of fission events that should be biased must be upwards of 80% (Fig 2D).

Interestingly, even when the same experimental distribution of DUMP(s) was implemented in simulations with no biased fission, the baseline probability for DUMP inheritance was around 50% (Fig 2D). This implies that spatial organization of DUMPs within the mitochondrial network may also play a crucial role in their inheritance. Biased fission might contribute to altering mitochondrial structure in such a way that DUMPs are not situated on the most elongated mitochondrial branch, which is typically destined for inheritance (Fig 1G and 1H).

We next addressed how biased fission improves the asymmetric retention of DUMPs. Our model simulations showed that biased fission led to the formation of smaller DUMP-containing mitochondria with increased diffusive mobility (S1I and S6A Figs). These smaller structures would undergo fusion and biased fission with nearby mitochondria networks until the DUMP-containing regions fused together. While most mitochondrial fission events appeared biased adjacent to DUMP (Fig 3I), random fission placements still took place *in-vivo*. By adjusting the percentage of these biased fission events via increasing the number of random fission placements, we can assess how making fission placement less biased impacts the inheritance outcome.

To assess the impact of alterations in fission/fusion dynamics, diffusive motion, and biased fission on the temporal clustering of individual DUMP particles, we modified how DUMPs were seeded into the simulation at t = 1800s. We randomly assigned six non-adjacent

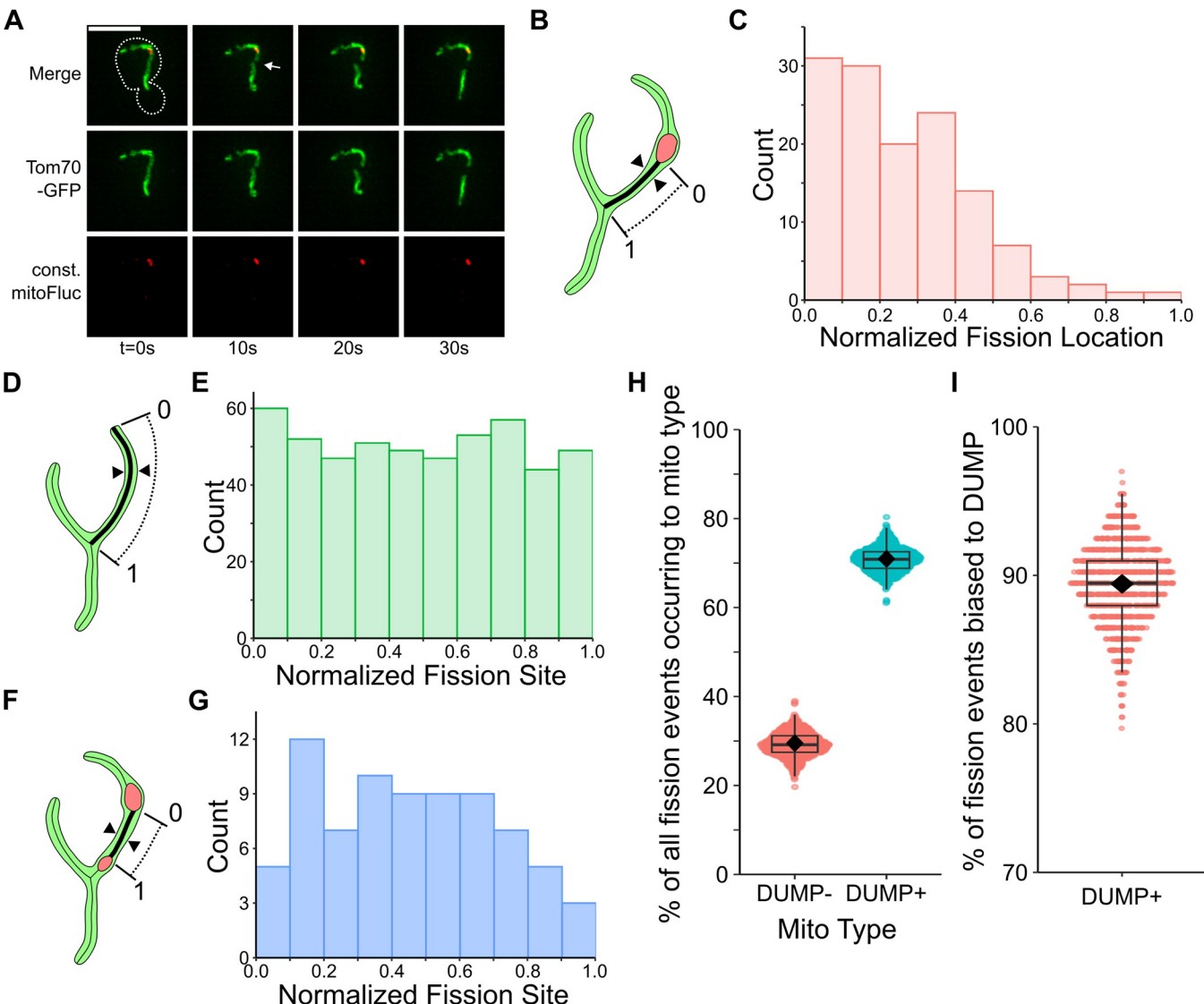

**Fig 3. Mitochondria fission occurs preferentially close to DUMPs in DUMP⁺ mitochondria.** (A) Example timeseries showing preferential mitochondria fission close to DUMP. White arrow indicates site of future fission. (B) Diagram showing fission position normalization procedure. Fission position is normalized against distance between DUMP surface boundary (defined as 0) to either closest mitochondrial tip or branch point (1), with fission site between. Arrow marks indicate where fission occurs; Black line marks mitochondria skeleton, with dotted projection indicating the length of mitochondria measured. (C) Normalized fission site preference towards DUMP where fission site is not flanked by another DUMP. n = 42 cells, 133 fission events observed. ****p = $1.224 \times 10^{-7}$, two sample Kolmogorov-Smirnov (KS) test against random uniform distribution with n = 133. (D) Diagram showing fission position normalization procedure in WT mitochondria. The tip or branch point is defined as 0, with the other opposing end defined as 1. Fission site is marked by double arrows; Bold black line marks mitochondria skeleton, with dotted projection indicating the length of mitochondria measured. (E) Normalized fission localization in WT cells. n = 151 cells, 509 fission events observed. Two sample KS test against random uniform distribution with n = 509, NS p = 0.3017. (F) Diagram showing fission position normalization procedure when fission site occurs between two DUMPs. The boundary of a randomly selected DUMP was defined as 0, with the other defined as 1. Fission site is marked by double arrows; Bold black line marks mitochondria skeleton, with dotted projection indicating the length of mitochondria measured. (G) Fission preference towards DUMP where fission site is flanked by DUMP. n = 45 cells, 162 fission events observed. Two sample KS test against random uniform distribution with n = 162, p = 0.6795. (H) Bootstrapped percentage of fission events affecting each mitochondria type in mitoFluc cells. n = 295 fission events. (I) Bootstrapped percentage of fission events affecting DUMP⁺ mitochondria that are biased. Biased fission is defined as any fission event that has a normalized distance of <0.5, as depicted in (C), n = 209 fission events. (H, I) n = 62 cells represented, bootstrap iterations = 1000, sample size = n fission events.

mitochondrial units as the DUMP-containing units, and then continued the simulation to monitor their migration and clustering patterns over time. Over the course of the simulations, individual DUMP particles gradually clustered into fewer DUMP clusters over time, which concurrently reduced their likelihood of inheritance (Fig 2E and 2H). This is in stark contrast to the case without biased fission, where the number of DUMP clusters did not decrease over time (Figs 2E, S6B and S6E). Changes in fission/fusion, diffusive motion parameters similarly had no impact on DUMP clustering over time (Fig 2F, 2G, 2I and 2J). Additionally, alterations in fission/fusion, diffusive motion parameters with biased fission had no additional synergistic improvement on DUMP clustering or inheritance (S6C, S6D, S6F and S6G Fig). Additionally, setting biased fission not immediately adjacent to DUMP particles still resulted in clustering of DUMPs over time in our model, with also a reduction in probability of inheritance (S7 Fig). These model results suggest that biased fission facilitates DUMP clustering which greatly diminishes the chance that the bud inherits DUMP(s).

### Biased fission facilitates DUMP clustering *in vivo*

To experimentally characterize the spatial pattern of mitochondrial fission with respect to DUMP location, we utilized high resolution spatiotemporal imaging to characterize where fission occurs on DUMP$^+$ mitochondria. Our observations indicated that mitochondria fission frequently occurs close to DUMP (Fig 3B and 3C). In contrast, the location of fission events was evenly dispersed along the entire length of the branch in wild type (WT) mitochondria undergoing fission. (Fig 3D and 3E). When the fission site was flanked by DUMP on both sides, fission placement was also random (Figs 3F and 3G, S8A). Additionally, DUMP$^+$ mitochondria experienced more fission events compared to DUMP$^-$ mitochondria (Figs 1D and 3H), with ~89% of DUMP$^+$ fission events occurring near DUMP (Fig 3I). Taken together, our experimental data strongly indicates that not only do most fission events occur to DUMP + mitochondria, but also are biased close to DUMP.

Unlike fission, the placement of mitochondrial fusion was not influenced by the presence of DUMP. In WT mitochondria, fusion events occurred mostly at the tips of the mitochondria (S8B–S8D Fig). This pattern was preserved in mitoFluc-expressing cells, regardless of whether DUMP was present in either of the fusing mitochondria (S8E–S8G, S8I and S8J Fig). Even when DUMP was considered as a spatial landmark for normalizing the fusion site placement, we observed a similar pattern (S8H and S8K Fig), which implies that DUMP is localized at the tips of mitochondria.

To experimentally determine whether and how biased fission may facilitate DUMP clustering, we needed to initially introduce multiple DUMPs into the cells. To do so, we utilized a β-estradiol inducible system to control the expression of mitoFluc [24]. Using a constitutive promoter (GAP) resulted in a steady-state average of 1–2 DUMPs per cell, in which the DUMP is presumably already clustered. To better visualize the DUMP clustering process, we resorted to the inducible expression of mitoFluc that led to the initial formation of many DUMPs randomly throughout the mitochondria network (Fig 4A and 4B). Analysis of DUMP localization in cells with 90-minute mitoFluc induction showed that the probability of aggregate inheritance increased with the number of discrete aggregate bodies (Fig 4C). This finding aligns with our model's prediction which predicted a similar trend when we tested simulations with an increasing number of DUMP clusters, while maintaining the overall DUMP volume equivalent to six particles. With multiple DUMPs present, biased fission was predicted to isolate individual DUMP$^+$ mitochondria that eventually fused together, resulting in fewer but larger clusters of DUMPs. Consistent with this, we also observed that when multiple DUMPs were present, biased fission isolated individual DUMP+ mitochondria which eventually fused together (Fig 4E).

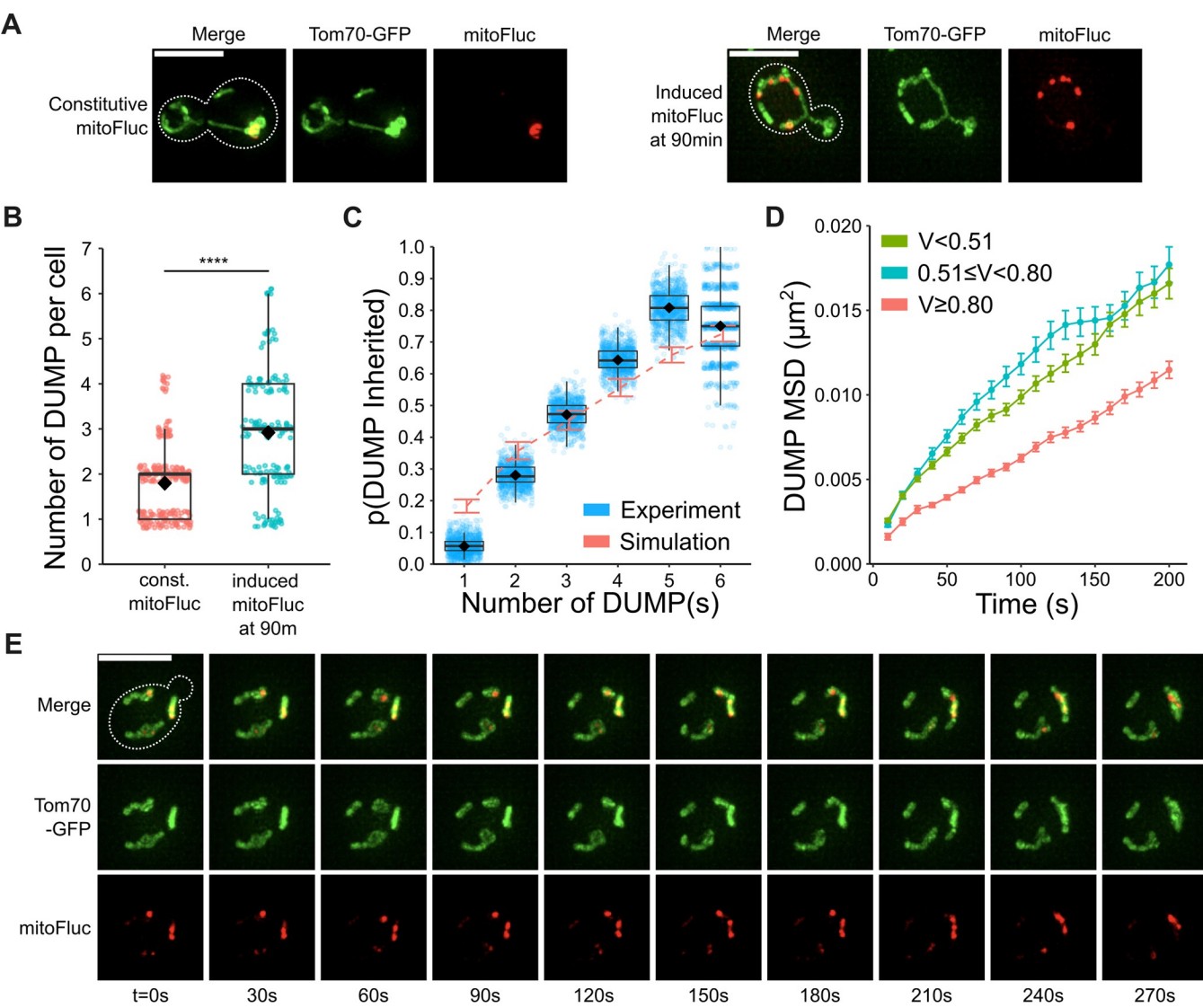

**Fig 4. In vivo DUMP inheritance increases with number of DUMPs; biased fission appears to aid in clustering DUMPs.** (A) Representative snapshots of cells constitutively expressing mitoFluc (left), compared cells with induced mitoFluc expression after exposure for 90min of 1μM β-Estradiol (right). (B) Number of DUMP(s) after 90min of 1μM β-Estradiol exposure. Mann Whitney U test ****p<0.0001. (C) Probability of DUMP inheritance vs total number of DUMP clusters in cell; probabilities are derived from bootstrap sampling from all cells with n DUMP with iterations = 1000, sampling size = number of cells with n DUMP, where number of cells = (70, 170, 184, 134, 52, 16) for n = 1–6 respectively. Red dashed line and error bars mark mean and range generated from simulation with mitoFluc parameters set, n = 200 simulations for each tested number of DUMP, bootstrapped with iterations = 1000, sampling size = 200 (Table 2). (D) DUMP MSD relative to mitochondria stratified by volume (V) tercile. n = 119 cells with n≥472, 508, 537 trajectories for DUMPs of V<0.51, 0.51≤V<0.80, V≥0.80 μm³ respectively. D = $9.07\times10^{-5}\pm0.27\times10^{-5}$, $9.99\times10^{-5}\pm0.37\times10^{-5}$, $5.97\times10^{-5}\pm0.13\times10^{-5}$ μm²/s respectively, with all **** p<0.0001, h₀: D = 0. Mean±SEM. (E) Representative timeseries of DUMP+ mitochondria undergoing targeted fission, followed by fusion that results in two separate DUMPs fusing into one larger DUMP.

This raised the question of whether mitochondrial remodeling is necessary for DUMP clustering since diffusion of protein aggregates within the mitochondria matrix could also lead to clustering. Interestingly, an analysis of our *in-vivo* high spatiotemporal resolution movies revealed that the relative diffusion of DUMP within the mitochondrial matrix was slow (~$1\times10^{-4}$ μm²/s, Fig 4D), even slower than the diffusive motion of the mitochondria themselves (~$8\times10^{-4}$ μm²/s, Fig 1F). The average nearest neighbor distance between two DUMPs in the same mitochondria was measured to be ~1.64±0.96 μm. Assuming they are both in the

same continuous mitochondrial matrix, two DUMPs would take an impractically long time, about $t = \frac{x^2}{2D} \sim 3.7$ hours, to meet and cluster via one dimensional diffusion within the matrix. This duration even surpasses the time typically needed for ~2.5 cell cycles (3.7hrs/90 min) in budding yeast [63]. Moreover, larger DUMPs exhibit reduced mobility within the matrix (Fig 4D). This suggests that as DUMPs grow larger, either through mitoFluc aggregation or clustering, their ability to cluster further via matrix diffusion is curtailed. Therefore, this analysis supports the concept that mitochondria remodeling, specifically biased fission, is crucial for expediting DUMP clustering and ensuring asymmetric DUMP retention.

## Fission proteins Dnm1 and Mdv1 preferentially localize to the vicinity of DUMP

To further understand how biased fission occurs at a molecular level, we tested the possibility that the outer membrane (OM) fission machinery is preferentially localized to DUMP proximate regions. We labeled two components of the OM fission machinery, Dnm1 and Mdv1 [66,67], with GFP N-terminally to quantify their surface distribution on WT, DUMP⁻, and DUMP⁺ mitochondria. In WT cells, both Dnm1 and Mdv1 colocalize and form uniformly spaced punctate structures throughout the mitochondria network (Figs 5A and S9A) [68]. However, in mitoFluc cells we observed an increased concentration of Dnm1 and Mdv1 surface density specifically where DUMP was located (Figs 5B and S9B), with lower surface density on other parts of the DUMP⁺ and DUMP⁻ mitochondria (Figs 5E, 5F and S9E). Importantly, this increase was not the result of an increase in the total number of Dnm1 or Mdv1 puncta per cell (Figs 5G and S9F).

The localized concentration of fission proteins around DUMP implies a mechanism of recruitment. Since Mdv1 is a peripheral membrane protein that serves as an adaptor for Dnm1, we tested how the deletion of either *MDV1* or *DNM1* influences the localization of their counterpart. In *Δmdv1* cells expressing GFP-Dnm1, we observed a Dnm1 surface density enrichment near DUMP similar to that in *MDV1* mitoFluc cells (Fig 5C, 5D and 5H). Similarly, the deletion of *CAF4*, a gene encoding an Mdv1 paralog [69] that does not form puncta (S9H–S9I Fig), did not alter Dnm1 preferential localization (S9G–S9I Fig). However, in *Δdnm1* cells, GFP-Mdv1 could not form puncta on mitochondria, resulting in a faint GFP signal mirroring the mitochondria structure with no apparent GFP-Mdv1 bias near DUMP (S9C and S9D Fig). Additionally, only Dnm1 and Mdv1 showed preferential localization adjacent to DUMP, while Fis1 and Caf4 exhibited no discernible localization bias (S10A–S10D Fig). Deletion of the fission machinery responsible for Dnm1 recruitment (*Δfis1*, *Δcaf4*, *Δmdv1Δcaf4*, *Δmdv1Δfis1*) led to disassociation of Dnm1 from the mitochondria (S10E–S10F and S11 Figs). These observations suggest that Dnm1 is the primary fission protein responsible for biasing association of the fission machinery towards DUMP proximate areas.

## Mitochondria fission and fusion dynamics are required for DUMP consolidation and asymmetric inheritance

We next perturbed the mitochondrial fission machineries. Deletion of *DNM1* leads to the formation of a distinct lattice-like mitochondrial structures, due to an imbalance between fusion and fission that results in mitochondria hyper-fusion (S12A and S12B Fig) [44,70]. Intriguingly, upon induction DUMPs formed clusters primarily in these lattice-like regions (S12C and S12D Fig) with an improved DUMP inheritance compared to cells with intact *DNM1* (S12E and S12F Fig). Although these lattice-like regions constitute most of the cell's total mitochondrial volume (S12G and S12H Fig), they alone cannot justify the preferential formation of DUMP(s) (Pearson point-biserial correlation, p = 0.1163, $h_0$: no relationship between lattice

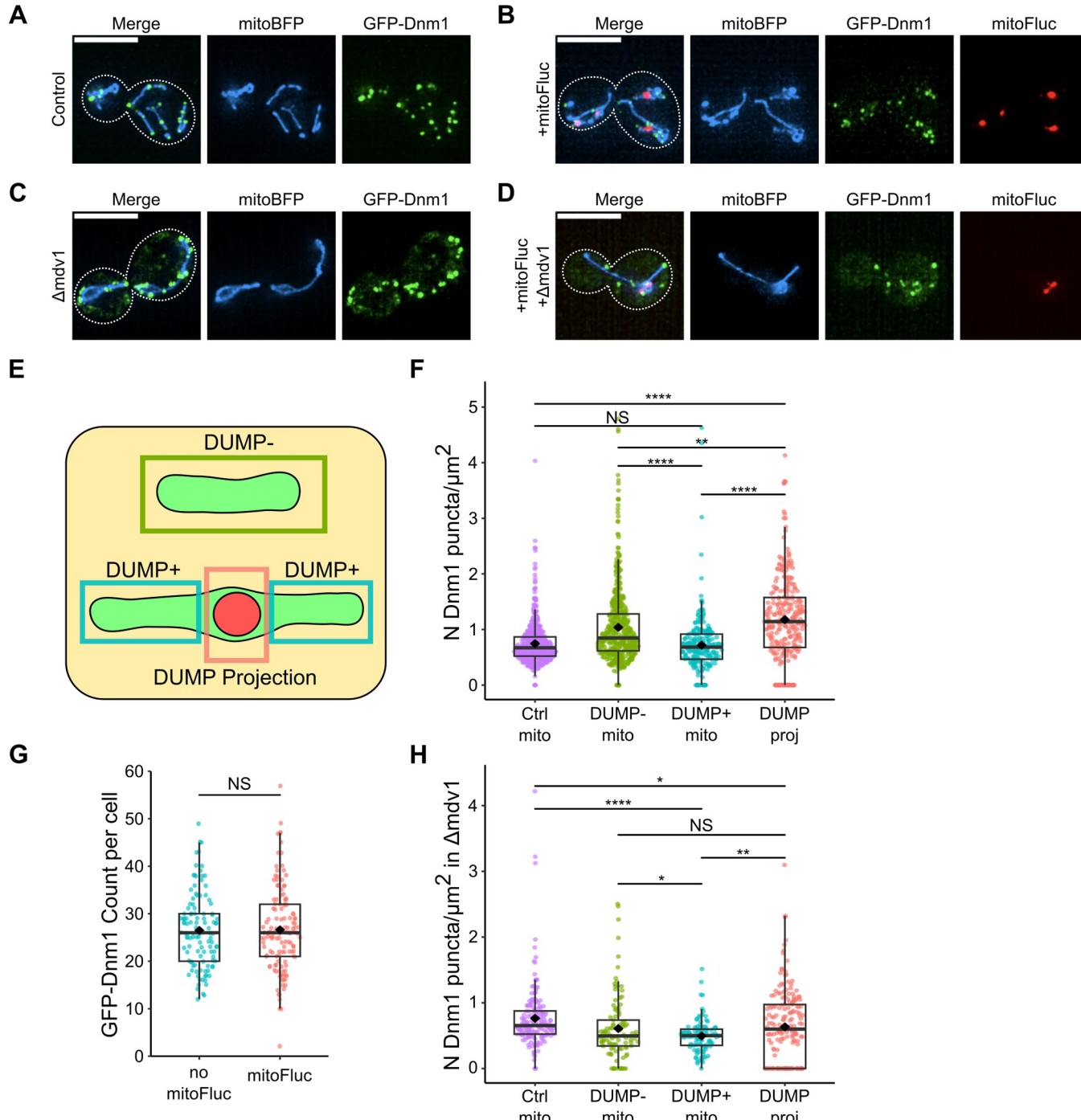

**Fig 5. Dnm1 exhibits preferential localization next to DUMP(s).** Representative GFP-Dnm1 localization in non-mitoFluc cells (A) and mitoFluc cells (B). Representative GFP-Dnm1 localization in Δmdv1 cells without (C) and with (D) mitoFluc expression. (E) Quantification scheme for determination of GFP-puncta density used in (F, H). DUMP is shown as a red circle in a green mitochondria. DUMP$^+$ mitochondria are subclassified into non-DUMP containing sections of the mitochondria (DUMP$^+$) vs. the local region of the mitochondria containing DUMP (DUMP projection). See methods for more details. Figure not drawn to scale. (F, H) Surface density measurements of punctate structures on mitochondria regions. Each point represents the surface density measurement on a specific region of a mitochondrion. (F) Dnm1 puncta surface density. Control = mitoBFP, GFP-Dnm1 cells; n = 114 cells with 561 mitochondria total. n = 138 cells for mitoFluc, with 770 mitochondria total, 524 which were DUMP$^-$, 246 DUMP$^+$; n = 298 DUMPs were found in DUMP$^+$ mitochondria, which were used to evaluate DUMP projection GFP-Dnm1 surface densities. Mann Whitney U test NS p = 0.341; ***p<0.001; ****p<0.0001. (G) Total number of GFP-Dnm1 puncta per cell with no-mitoFluc vs constitutive mitoFluc expression, n = 114, 138 cells respectively. Mann Whitney U test, p = 0.981. (H) Dnm1 puncta surface density with Δmdv1. Control = mitoBFP, GFP-Dnm1, Δmdv1 cells, n = 103 cells with 173 mitochondria total. n = 101 cells for mitoFluc, with 215 mitochondria total, 109 which were DUMP$^-$, 106 DUMP$^+$; n = 180 DUMPs were found in DUMP$^+$ mitochondria, which were used to

evaluate DUMP projection GFP-Dnm1 surface densities. Mann Whitney U test *0.01<p<0.05; ****p<0.0001. DUMP⁻ vs DUMP⁺ mitochondria NS p = 0.363; DUMP⁻ mitochondria vs DUMP projection NS p = 0.462.

size and DUMP location). Considering that DUMP formation in tubular mitochondria (with *DNM1*) occurs randomly throughout the mitochondrial structure and that lattice-like structures are not observed, we cannot attribute the formation of DUMP within lattice-like structures as a dominant mechanism for directing DUMP clustering. Given that the deletion of *DNM1* appears to excessively disrupt the system, we concluded that this model strain might not effectively represent the actual biological processes at play.

To preserve mitochondria tubulated structures, we made double gene deletion for both *DNM1* and *FZO1*. Notably, this double mutant is observed to preserve a WT-like tubulated mitochondria morphology [43,54,62]. We next used the inducible mitoFluc construct to randomly generate multiple DUMPs in *Δdnm1 Δfzo1* mutant cells. Post induction, small DUMPs formed randomly throughout the mitochondria, then gradually coalesced into fewer, larger DUMPs (Fig 6A). However, in contrast to WT mitochondria, this process was slower in clustering DUMPs over time (Fig 7A). While DUMPs could diffuse in the matrix of both WT and *Δdnm1 Δfzo1* mitochondria, *Δdnm1 Δfzo1* mitochondria were unable to undergo fission (S13A and S13B Fig). The difference in DUMP clustering speed between WT and *Δdnm1 Δfzo1* mitochondria suggests that biased fission accelerates the rate at which DUMPs cluster. However, for DUMPs to cluster only via matrix migration, a continuous matrix that serves as a conduit for DUMPs to traverse is required. To test this, we examined the combined deletion of *DNM1* and *MGM1*, the latter encoding a protein required for tethering and fusing inner mitochondrial (IM) membranes [71–73]. Although *MGM1* deletion alone results in mitochondria fragmentation, its combined deletion with *Δdnm1* yields tubulated mitochondria exhibiting a relatively WT network-like morphology [70]. However, the inability to fuse the IM disrupts the continuity of mitochondria matrix [74].

As expected, inducing mitoFluc expression in *Δdnm1 Δmgm1* cells culminated in the formation of multiple DUMPs throughout the mitochondria network that were unable to cluster (Fig 6B). Most importantly, we observed that over the course of 90 minutes, the number of DUMP clusters reduced drastically for only the WT strain, whereas the number of DUMP clusters showed minimal to no change for this double mutant (Fig 7A). The Δdnm1 Δmgm1 cell line displayed the poorest performance in terms of DUMP clustering over time. This was closely followed by the Δdnm1 Δfzo1 cells, with the wild-type (WT) cells exhibiting the most efficient DUMP clustering process (Fig 7A). Additionally, the dispersion of DUMPs in *Δdnm1Δmgm1* cells was greater than all other strains (Fig 7B). Consequently, the likelihood of DUMP inheritance by the bud in this double mutant exceeded that of WT and all other mutant strains tested (Fig 7C). Given that the rates of mitochondria fission and fusion were also diminished to zero in *Δdnm1*, *Δdnm1Δfzo1*, *Δdnm1Δmgm1* (S13 Fig), these results suggest that in the absence of fission and fusion dynamics, DUMPs cluster together at a reduced rate, thus resulting in a greater number of DUMPs and an increased rate of failure in asymmetric inheritance. This pattern remained consistent even when DUMP was constitutively expressed, with increased DUMP number and decreased clustering resulting in a greater likelihood for inheritance by the bud (Fig 7D–7F).

## Discussion

In this study, we aimed to elucidate the influence of mitochondrial dynamics on the asymmetric inheritance of protein aggregates-containing mitochondria. By engaging an experimentally parameterized in-silico model of mitochondria segregation with experimental investigation,

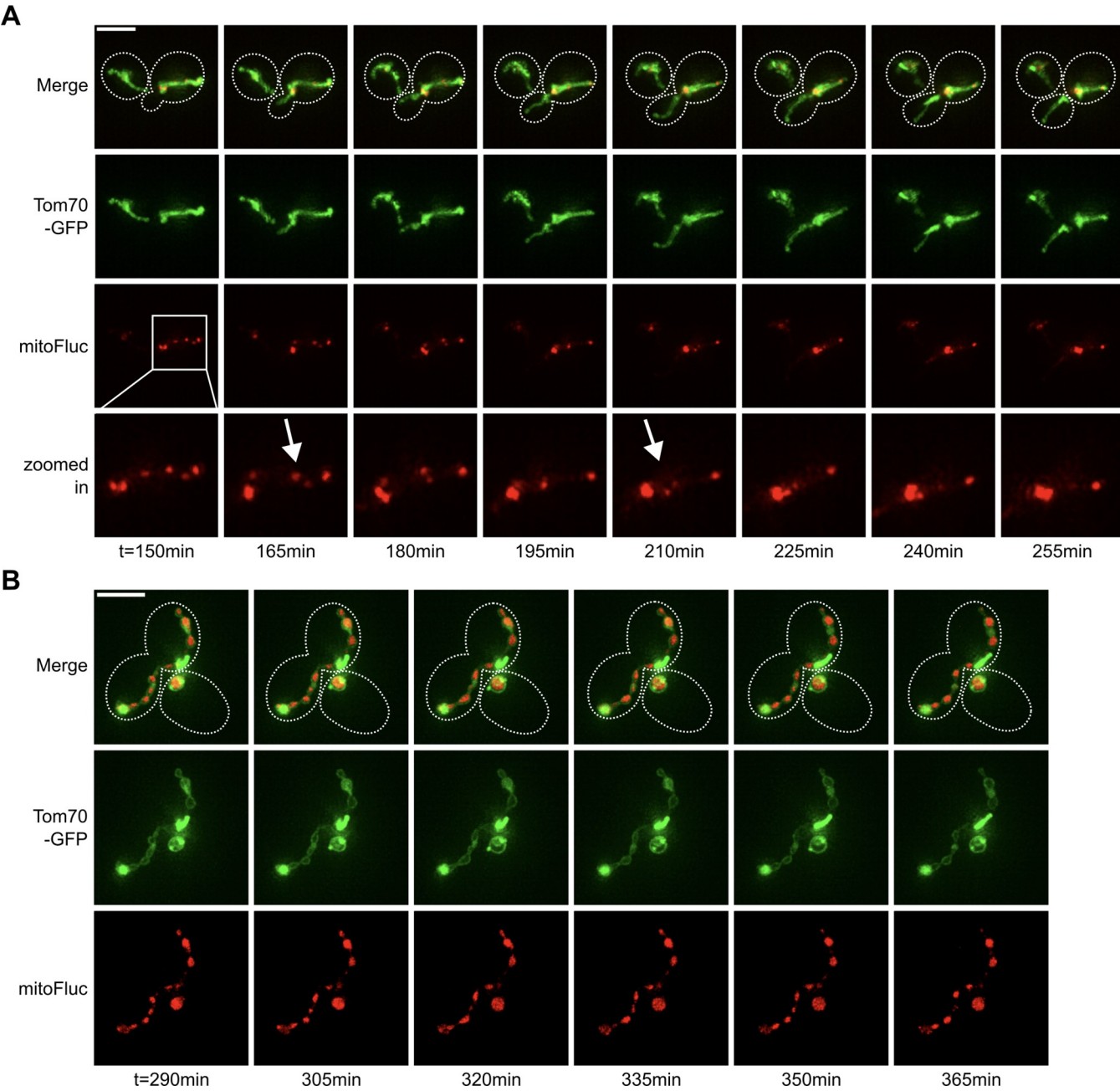

**Fig 6. DUMP diffusion in the mitochondria is slow and limited by deletion of inner-membrane fusion machinery.** (A) Timeseries of Δdnm1Δfzo1 cell with inducible mitoFluc with 1μm β-Estradiol. White arrows indicate DUMP fusion events. (B) Timeseries of Δdnm1Δmgm1 cell with inducible mitoFluc with 1μM β-Estradiol.

we uncovered that biased fission aids the clustering of multiple DUMPs into fewer though larger clusters, thereby promoting asymmetric retention in the progenitor cell. By manipulating gene deletion mutants that affect various components of inner and outer membrane fission or fusion machinery, we established that DUMP diffusion within mitochondrial matrix is inefficient for clustering, and that biased fission operates as the primary catalyst.

Our findings highlight that one outcome of mitochondrial self-organization is its capability to minimize the number of DUMPs through clustering, which consequently enhances

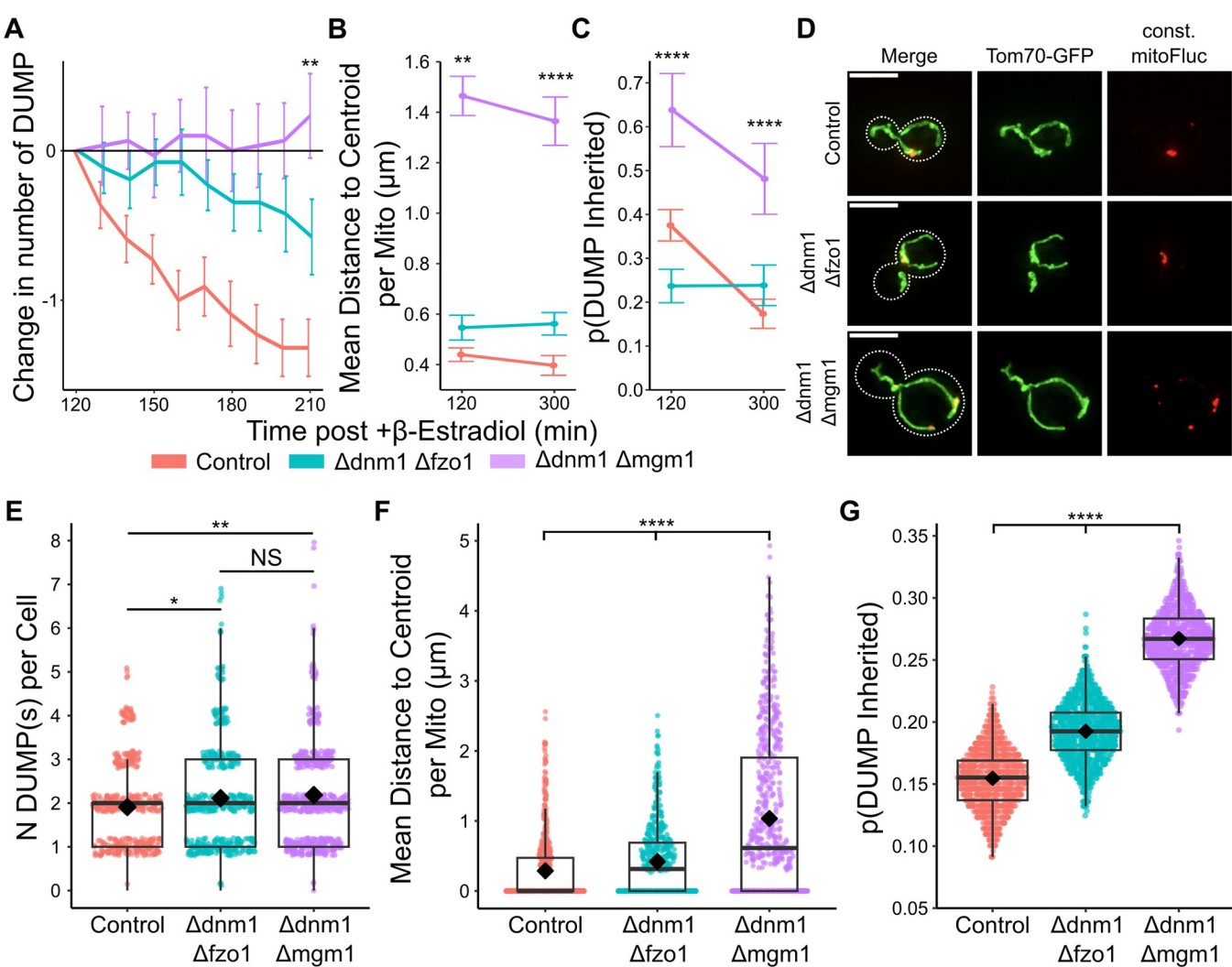

**Fig 7. Loss of fission machinery reduces DUMP clustering and increases the likelihood of DUMP inheritance.** (A-C) Induced mitoFluc cells over time at 120min and 210min post β-Estradiol induction. Color bar is below. (A) Change in number of DUMP over time in individual cells, n = 22, 26, 31 cells for control, Δdnm1Δfzo1, Δdnm1Δmgm1 respectively. **p<0.01 for all comparisons (B) Mean DUMP distance to centroid within each mitochondria. n = 323, 123, 190 cells for control, Δdnm1Δfzo1, Δdnm1Δmgm1 at 120min respectively (**p<0.01 for all comparisons), and n = 158, 202, 143 cells for at 300min (****p<0.0001 for all comparisons). (C) Bootstrapped probability of DUMP inheritance with iterations = 1000, sample size = number of cells identical to (B). ****p<0.0001 for all comparisons for both t = 120, 300 min. (D) Representative snapshots of control and deletion strain cells constitutively expressing mitoFluc. (E-G) Constitutive mitoFluc cells (E) Number of DUMPs per cell in cells expressing mitoFluc constitutively. Mann Whitney U test; **p<0.01, *p<0.05, NS p = 0.558. (F) Mean DUMP distance to centroid per mitochondria. n = 553, 513, 608 mitochondria for control, Δdnm1Δfzo1, Δdnm1Δmgm1 respectively (G) Bootstrapped probability of DUMP inheritance with iterations = 1000, sampling size = n; n = 219, 265, 367 cells for WT, Δdnm1Δfzo1, Δdnm1Δmgm1 respectively. (F, G) Mann Whitney U test, ****p<0.0001.

asymmetric retention of DUMP. Biased fission may define a common theme that enables the mitochondria to isolate and consolidate its dysfunctional parts to cope with cellular injuries. While yeast cells utilize biased fission of mitochondria to facilitate asymmetric DUMP inheritance, mammalian cells appear to employ a similar organizational strategy to selectively degrade the aggregate-containing mitochondria. In mitochondria containing aggregates, sub-domains containing these aggregates after fission have been observed to recruit increased Parkin [75], a promoter of mitophagy [76,77]. While the final outcomes—inheritance in our study vs. mitophagy in mammalian cells—are distinct, the strategy of using fission constructively to

isolate dysfunctional mitochondria could be a common theme. Interestingly, alpha-synuclein, a principal component of Lewy bodies in synucleinopathies and detected both surrounding and inside mitochondria [78,79], can stimulate an increased translocation of Drp1 (a mammalian Dnm1 homolog [80]) to the mitochondria [81,82] with ensuing mitochondrial fragmentation [83]. This may suggest a comparable mechanism whereby excess alpha-synuclein associated with mitochondrial triggers biased fission analogous to our observation in yeast. In addition, prior studies showed that dysfunctional mitochondrial subdomains, exhibiting low membrane potential or high levels of reactive oxygen species, are sequestered at the mitochondrial tips, subsequently undergo fission [36]. However, the precise mechanisms through which these compromised subdomains are confined to the mitochondrial tips, and the specific role of fission and/or fusion in facilitating this organization, remain elusive.

While our study observed the concentration of Dnm1 around DUMPs, we currently lack an understanding of this localized recruitment. While we speculate that Dnm1 could be recruited to DUMPs through curvature sensing, an important property of dynamin proteins [84–87], the precise domains responsible for curvature generation and sensing in Dnm1 remain to be identified. This is especially pertinent given that the regions of mitochondria where DUMPs are situated exhibit a larger diameter than tubulated WT, DUMP$^-$, or even non-DUMP-containing regions of DUMP$^+$ mitochondria. Concentration of Dnm1 around DUMP might be a result of local curvature in the mitochondria membrane induced by the DUMP. Alternatively, local expansion of the mitochondria membrane around a DUMP might lead to increased local membrane tension. Membrane tension is essential for the action of dynamin family proteins to break membranes [88], with high membrane tension enhancing fission efficiency [89–91]. The increased membrane tension and the potential curvature-induced recruitment of Dnm1 might synergistically prejudice fission placement.

Additionally, while our model was designed to examine the organization of mitochondrial network in response to DUMP accumulation and the subsequent influence on asymmetric DUMP retention, it was inadequate for recapitulating non-tubular mitochondria. As evident in the *Δdnm1* mutant, mitochondria adopted lattice-like structures that acted as preferential sites for DUMP accumulation. The specific mechanism driving the formation of these lattice-like regions remains largely unclear, extending only to a broad association with imbalanced fission vs fusion. When juxtaposed with our observations of DUMP(s) diffusing through the mitochondrial matrix, it raises an intriguing possibility that complex mitochondrial structures can influence location of damage accumulation even in the absence of fission.

Another intriguing finding of our study is the role of Mgm1 in facilitating the consolidation of DUMPs within the matrix. In the absence of Mgm1, DUMPs remain dispersed rather than clustered, underscoring the significance of inner membrane dynamics in clustering. Interestingly, Mgm1 is the yeast counterpart of the human OPA1 gene, which when mutated, has implications in dominant optic atrophy (DOA), a form of neurodegenerative disease [92–94]. Both Mgm1 and OPA1 are central to mitochondrial IM fusion and cristae remodeling [71,72]. While the root cause of mitochondria dysfunction in DOA is not well characterized, mutation or deletion of OPA1 can induce mitochondria dysfunction [74,95]. Our findings hint that changes in IM remodeling dynamics can contribute to the mitigation and containment of mitochondrial damage, especially under proteotoxic stress. A potential underlying mechanism damage consolidation could be the remodeling of the matrix cristae, which might facilitate DUMP migration within the IM possibly through peristaltic-like motions. Further studies are needed to shed light on the nuanced interplay between inner and outer membrane fission and fusion events that lead to DUMP clustering and isolation. Such insights would also help ascertain if this mechanism applies universally to other forms of mitochondrial dysfunction.

## Materials and methods

### Plasmids

Plasmids used in this study are listed in Table 3. Molecular cloning was performed by Gibson Assembly (NEB E5510S). Q5 high-fidelity DNA polymerase was used for all PCR reactions (NEB M0491L).

For N-terminal tagging of yeast proteins with GFP, the gene of interest was cloned with its native promoter and terminator from a WT (wild-type) strain with primers designed from the open reading frame (ORF) insert from the molecular barcoded yeast open reading frame (MoBY-ORF) library [97,98]. Primers had homology to pRS316 such that the gene ORF insert was able to be Gibson assembled into pRS316 digested by EcoRI (NEB R3101). GFP was isolated from pFA6A-link-yoEGFP-SpHis5 (Addgene 44836) using PCR and Gibson assembled into pRS316 containing the gene of interest.

To generate the non-mutant luciferase, the FlucSM construct (RLB1119), which originally had the R188Q mutation, was reverted to the wild type (Q to R) (RLB1123) using NEB Q5 site-directed mutagenesis (E0554S). All plasmid constructs were confirmed through Sanger sequencing.

### Yeast strains

All strains were grown in synthetic-complete (SC) (Sunrise Biosciences, 1459–100) unless otherwise noted. Media contained 2% glucose (SC-Complete). Optical density at 600nm was used to estimate the density of yeast cells used in various experiments.

Yeast strains used in this study were based on the BY4741 strain background (MATa his3Δ1 leu2Δ0 met15Δ0 ura3Δ0) and are listed in Table 4. Gene deletion and C-terminal fluorescent protein tagging were performed with polymerase chain reaction (PCR)-mediated homologous recombination [99] and verified by PCR genotyping. Integration of inducible or constitutive mitoFluc was always the last transformation if multiple deletions/tagging were needed (i.e. for Δdnm1Δmgm1 with mitoFluc, deletion of both dnm1 and mgm1 preceded mitoFluc integration).

For N-terminal fluorescent protein tagging, a plasmid containing the ORF of the protein of interest was constructed (see Plasmids, Table 3) and GFP integrated at the N-terminus using Gibson assembly. The GFP-tagged sequence on the plasmid was then cloned with primers

**Table 3. Plasmids used in this study.**

| Plasmid ID | Source | Contents |
|---|---|---|
| RLB1110 | Sikorski et al. 1989 [96] | pRS313 |
| RLB1111 | Sikorski et al. 1989 [96] | pRS316 |
| RLB1112 | This Paper | pRS316 MobyDnm1 |
| RLB1113 | This Paper | pRS316 MobyGFPDnm1 |
| RLB1114 | This Paper | pRS316 MobyMdv1 |
| RLB1115 | This Paper | pRS316 MobyGFPMdv1 |
| RLB1116 | Addgene 44836 | pFA6A-link-yoEGFP-SpHis5 |
| RLB1117 | Addgene 112050 | pJW1662 |
| RLB1118 | Ruan et al. 2020 [24] | GEM-pGAL1-MTS-FlucSM-mCherry |
| RLB1119 | Ruan et al. 2020 [24] | TRP1::pGAP-MTS-FlucSM-mCherry |
| RLB1120 | Addgene 49151 | mitoBFP |
| RLB1122 | This Paper | TRP1::pGAP-MTS-Fluc (WT)-mCherry (wild type luciferase) |
| RLB1123 | This Paper | TRP1::pGAP-MTS-mCherry |

**Table 4. Yeast strains used in study.**

| Strain ID | Genotype | Isogenic to |
|---|---|---|
| RLY10250 | BY4741 MATa (Wild Type) | N/A |
| RLY10251 | BY4741 MATa trp1::MTS-mCherry:Nrsr | RLY10250 |
| RLY10252 | BY4741 MATa trp1::MTS-mCherry:Nrsr; Δdnm1::KanMX | RLY10250 |
| RLY10258 | BY4741 MATa Tom70-GFP:His3MX | RLY10250 |
| RLY10261 | BY4741 MATa Tom70-GFP:His3MX; trp1::MTS-FlucSM-mCherry:Nrsr | RLY10258 |
| RLY10260 | BY4741 MATa Tom70-GFP:His3MX; chr4::GEM-pGAL1-MTS-FlucSM-mCherry:KanMX | RLY10258 |
| RLY10259 | BY4741 MATa Tom70-GFP:His3MX; Δdnm1::URA3 | RLY10258 |
| RLY10268 | BY4741 MATa Tom70-GFP:His3MX; Δdnm1::URA3; trp1::MTS-FlucSM-mCherry:Nrsr | RLY10259 |
| RLY10265 | BY4741 MATa Tom70-GFP:His3MX; Δdnm1::URA3; chr4::GEM-pGAL1-MTS-FlucSM-mCherry:KanMX | RLY10259 |
| RLY10262 | BY4741 MATa Tom70-GFP:His3MX; Δdnm1::URA3; Δmgm1::LEU2 | RLY10259 |
| RLY10264 | BY4741 MATa Tom70-GFP:His3MX; Δdnm1::URA3; Δmgm1::LEU2; chr4::GEM-pGAL1-MTS-FlucSM-mCherry:KanMX | RLY10262 |
| RLY10263 | BY4741 MATa Tom70-GFP:His3MX; Δdnm1::URA3; Δmgm1::LEU2; trp1::MTS-FlucSM-mCherry:Nrsr | RLY10262 |
| RLY10273 | BY4741 MATa Tom70-GFP:His3MX; Δdnm1::URA3; Δfzo1::LEU2 | RLY10259 |
| RLY10275 | BY4741 MATa Tom70-GFP:His3MX; Δdnm1::URA3; Δfzo1::LEU2; trp1::MTS-FlucSM-mCherry:Nrsr | RLY10273 |
| RLY10278 | BY4741 MATa Tom70-GFP:His3MX; Δdnm1::URA3; Δfzo1::LEU2; chr4::GEM-pGAL1-MTS-FlucSM-mCherry:KanMX | RLY10273 |
| RLY10255 | BY4741 MATa Δmdv1::URA3 | RLY10250 |
| RLY10254 | BY4741 MATa Δdnm1::URA3 | RLY10250 |
| RLY10257 | BY4741 MATa GFP-Mdv1 | RLY10255 |
| RLY10256 | BY4741 MATa GFP-Dnm1 | RLY10254 |
| RLY10274 | BY4741 MATa ura3Δ0::mitoBFP:His3MX | RLY10250 |
| RLY10279 | BY4741 MATa ura3Δ0::mitoBFP:His3MX; trp1::MTS-FlucSM-mCherry:Nrsr | RLY10274 |
| RLY10267 | BY4741 MATa ura3Δ0::mitoBFP:His3MX; GFP-Mdv1 | RLY10257 |
| RLY10266 | BY4741 MATa ura3Δ0::mitoBFP:His3MX; GFP-Dnm1 | RLY10256 |
| RLY10269 | BY4741 MATa ura3Δ0::mitoBFP:His3MX; GFP-Mdv1; Δdnm1::URA3 | RLY10267 |
| RLY10270 | BY4741 MATa ura3Δ0::mitoBFP:His3MX; GFP-Dnm1; Δmdv1::URA3 | RLY10266 |
| RLY10271 | BY4741 MATa ura3Δ0::mitoBFP:His3MX; GFP-Mdv1; trp1::MTS-FlucSM-mCherry:Nrsr | RLY10267 |
| RLY10272 | BY4741 MATa ura3Δ0::mitoBFP:His3MX; GFP-Dnm1; trp1::MTS-FlucSM-mCherry:Nrsr | RLY10266 |
| RLY10276 | BY4741 MATa ura3Δ0::mitoBFP:His3MX; GFP-Mdv1; Δdnm1::URA3; trp1::MTS-FlucSM-mCherry:Nrsr | RLY10269 |
| RLY10277 | BY4741 MATa ura3Δ0::mitoBFP:His3MX; GFP-Dnm1; Δmdv1::URA3; trp1::MTS-FlucSM-mCherry:Nrsr | RLY10270 |
| RLY10288 | BY4741 MATa ura3Δ0::mitoBFP:His3MX; Fis1-GFP:Leu2 | RLY10274 |
| RLY10287 | BY4741 MATa ura3Δ0::mitoBFP:His3MX; Caf4-GFP:Leu2 | RLY10274 |
| RLY10291 | BY4741 MATa ura3Δ0::mitoBFP:His3MX; Fis1-GFP:Leu2; trp1::MTS-FlucSM-mCherry:Nrsr | RLY10288 |
| RLY10290 | BY4741 MATa ura3Δ0::mitoBFP:His3MX; Caf4-GFP:Leu2; trp1::MTS-FlucSM-mCherry:Nrsr | RLY10287 |
| RLY10253 | By4741 MATa Tom70-GFP:His3MX; Δfzo1::Leu | RLY10258 |
| RLY10286 | BY4741 MATa ura3Δ0::mitoBFP:His3MX; GFP-Dnm1; Δmdv1::URA3; Δfis1::Leu; trp1::MTS-FlucSM-mCherry:Nrsr | RLY10283 |
| RLY10283 | BY4741 MATa ura3Δ0::mitoBFP:His3MX; GFP-Dnm1; Δmdv1::URA3; Δfis1::Leu | RLY10270 |
| RLY10280 | BY4741 MATa ura3Δ0::mitoBFP:His3MX; GFP-Dnm1; Δcaf4::Leu | RLY10266 |
| RLY10284 | BY4741 MATa ura3Δ0::mitoBFP:His3MX; GFP-Dnm1; Δcaf4::Leu; trp1::MTS-FlucSM-mCherry:Nrsr | RLY10280 |
| RLY10281 | BY4741 MATa ura3Δ0::mitoBFP:His3MX; GFP-Dnm1; Δmdv1::URA3; Δcaf4::Leu | RLY10270 |
| RLY10285 | BY4741 MATa ura3Δ0::mitoBFP:His3MX; GFP-Dnm1; Δmdv1::URA3; Δcaf4::Leu; trp1::MTS-FlucSM-mCherry:Nrsr | RLY10281 |
| RLY10282 | BY4741 MATa ura3Δ0::mitoBFP:His3MX; GFP-Dnm1; Δfis1::Leu | RLY10266 |
| RLY10289 | BY4741 MATa ura3Δ0::mitoBFP:His3MX; GFP-Dnm1; Δfis1::Leu; trp1::MTS-FlucSM-mCherry:Nrsr | RLY10282 |
| RLY10292 | BY4741 MATa trp1::MTS-Fluc-mCherry(wild type luciferase):Nrsr | RLY10250 |
| RLY10293 | BY4741 MATa Tom70-GFP:His3MX; trp1::MTS-Fluc-mCherry(wild type luciferase):Nrsr | RLY10258 |
| RLY10294 | BY4741 MATa Tom70-GFP:His3MX; trp1::MTS-mCherry:Nrsr | RLY10258 |

with homology to regions flanking the ORF insert from the MoBY-ORF library. Concurrently the gene of interest was deleted in a WT yeast strain via replacement with URA3. Cells were then transformed simultaneously with pRS313 and the GFP-protein PCR product and plated on SD-His plates. Colonies were then replica plated onto 5FOA plates, verified by replica plating on URA plates [100]. Colonies that grew on 5FOA but not URA plates were verified by PCR genotyping.

Yeast imaging conditions: Single colonies were inoculated into 5mL of SC-complete and grown in culture tubes tilted at 10 degrees running at 55 rpm at 30°C overnight for at minimum 15 hours (Fisherbrand 14-251-250). Overnight cultures were diluted to 0.05 OD600 in fresh SC complete media and grown for 3.5 hours at 30°C to logarithmic phase. For imaging, 500uL of the refreshed culture was spun down at 10000g for 30s and concentrated into approximately 100μL of residual media. This concentrated culture mix was then used for preparation of slides and dishes for imaging.

For visualization of mitochondria, Tom70 was C-terminally tagged with EGFP via homologous recombination (Addgene 44836). Mitochondria visualization with mitoBFP was only used (Addgene 49151) when the GFP channel was required for visualization of DiOC6 membrane potential staining or GFP-tagged fission machinery.

For introducing DUMP formation in mitochondria, we utilized a genetic construct, mitoFluc, from prior work [24,55]. MitoFluc consists of a mitochondrial targeting signal (MTS) conjugated to FlucSM (R188Q) and mCherry, constitutively expressed using a GAP promoter Inducible expression was achieved with the mitoFluc construct governed by an estradiol-activated GEM transcription factor, acting on a GAL1 promoter [101,102]. β-Estradiol was added to liquid cell cultures in SC-Complete at a final concentration of 1μM to induce expression, which would take approximately 90 minutes for DUMPs to appear in the majority of cells. For a negative control, an equivalent volume of ethanol (EtOH, 1μL/mL of culture) was added to liquid SC-Complete cultures.

Control experiments utilized MTS-mCherry (essentially mitoFluc minus the luciferase) and MTS-Fluc(WT)-mCherry (mitoFluc featuring the non-mutated luciferase form). These controls served to validate the specific effects and localization of the mitoFluc construct, ensuring that any observed phenomena were due to the presence of the unfolded proteins and not merely a result of the fluorescent markers or other inherent properties of the construct components.

## Confocal microscopy

Confocal movies were acquired with a Zeiss LSM 880 with Airyscan FAST microscope equipped with a 63x objective, 1.4 PlanApo oil.

For long term 3D time lapse microscopy (>10min), a 35mm glass bottomed dish (Mattek, P35G-0-14-C) was treated with 100μL of 1mg/μL Concavalin-A (ConA, Sigma L7647) for 10min. The ConA solution was then aspirated, and the dish was conditioned with 300uL of SC-Complete. 100μL of the concentrated cell culture mix was applied to the dish and incubated for 10min, followed by 5x washes with 1mL of fresh SC-Complete. An additional 4mL of fresh SC-complete was added to the dish followed by immediate imaging. Movies were recorded with incubation at 30°C and standard atmospheric conditions.

## Super-resolution and Deconvolved widefield microscopy

Structured illumination microscopy (SIM) images were acquired with a GE DeltaVision OMX-SR Super-Resolution Microscope 3D Structure Illumination equipped with high-sensitivity PCO scientific complementary metal-oxide semiconductor (sCMOS) cameras. Images

were acquired with a 60x1.42 NA UPlanApo oil objective (Cargille laser liquid 1.518RI). Images were acquired with a z-step interval of 125nm. GFP and mCherry were excited with 488nm and 568nm lasers respectively and collected with their standard GFP, RFP filter settings. Exposure time was 250ms for all channels, with 2% laser power for 488nm and 568nm lasers. DIC power was set at 10%. SIM images were reconstructed with Softworx and aligned following the Applied Precision protocols. All cells that were imaged using SIM were fixed in PFA prior to imaging.

For live cell images and movies, the same GE DeltaVision OMX-SR microscope was used, but set to conventional widefield settings in AcquireSR. Image snapshots were acquired with a z-step interval of 125nm, while short movies (<5min) with a z-step interval of 500nm, time interval step of 10 seconds. For snapshots, exposure was set to 100ms, 2% laser power for all channels, except for DIC which was set to 10% power. For movies, exposure settings were identical, laser power was set to 1.5% for all channels except for DIC which was maintained at 10% and 405nm at 5%. Images were deconvolved and aligned with Softworx following the Applied Precision protocols.

Prepared yeast culture 100μL concentrated culture mixes for imaging were vortexed and pipetted to resuspend the cell pellet and 1 μL was applied to 22x22mm No. 1.5 glass coverslips (VWR 48366–227) and topped with a microscope slide (VWR 16004–430). Slides and coverslips were kept in incubator at 30°C prior to use. Slides were used for a maximum of 15 minutes of imaging time before being replaced. Microscope environmental control was set to 30°C, standard atmospheric conditions.

## Membrane potential measurements

Yeast cells were cultured identically to that of yeast preparation for imaging, except 1mL of refreshed log-phase cell culture (in SC-Complete) was used for staining. Cells were washed once with staining buffer (10 mM HEPES, pH 7.6 + 5% glucose). After washing the cells once with staining buffer (10 mM Hepes, pH 7.6 + 5% glucose), we pelleted them via centrifugation and resuspended them in staining buffer with 21.875nM 3,3′-Dihexyloxacarbocyanine iodide (DiOC6, Invitrogen D273) at room temperature (~25°C) for 5 minutes. The cells were then washed two more times with staining buffer before being immediately subjected to microscopy. Using an image analysis pipeline, we segmented each mitochondrion and measured the average DiOC6 fluorescence intensity with each mitochondrion's associated voxels to obtain the corresponding fluorescence intensity. To account for differential dye uptake by individual cells, we normalized the mean mitochondria fluorescence intensity against cytoplasmic DiOC6 fluorescence intensity on a per-cell basis.

## Image analysis

We developed a high throughput image analysis pipeline for characterizing mitochondrial dynamics at high spatiotemporal single cell resolution [103]. Using our workflow, individual cells were automatically isolated from time lapse imaging (movies) or individual images. To ensure accurate quantitative analysis of fluorescence intensity, movies were corrected for photobleaching using an exponential fit. Histogram equalization was used to correct for bleaching for the purpose of mitochondria/aggregate segmentation.

In addition, the same pipeline was used to perform a range of other tasks, including (1) sorting individual cells based on whether or not they were visibly budding, (2) automatically identifying mother-bud regions of budding yeast cells and organelles associated with each region, (3) producing binary segmentations of mitochondria/aggregate structures, (4) tracking fission/fusion events of both mitochondria and aggregates in movies, (5) collecting structural

attributes such as volume and surface area of each segmented mitochondria/aggregate, and (6) quantifying organelle region specific attributes, including Dnm1 surface density.

For analysis, only cells that were visibly budding were used, while single cells were excluded. Each budding cell was divided into two regions, namely the mother (progenitor cell) and the bud (progeny), which were separated by the bud neck. We classified DUMPs localized in the bud region as inherited, as we did not observe de-novo mitoFluc formation in the bud region of budding cells until after cytokinesis.

## Quantification of rate of fission/fusion

Time-lapse z-stack movies were used to identify fission and fusion events in mitochondria. Movies were acquired at 10s per frame, for 5 minutes. From these movies, our pipeline recreated lineage maps of mitochondria fission/fusion, where mitochondria events were classified as no event, fission, or fusion. Mitochondria that exhibited no interaction with other mitochondria were classified as "no event", while mitochondria that underwent fission/fusion were classified as respectively. Due to the resolution limit for detection, mitochondria-mitochondria contact events were classified as fusion events as it is challenging to determine whether fusion occurred during these events. To calculate the probability rate of an event (fission/fusion/no event), for each frame F the event occurring in the transition to frame F+1 for each mitochondria was classified and counted, with the count normalized for the total number of events per movie and movie timestep.

To extrapolate rate of fission/fusion as a function of mitochondria volume ($K_{fis}$, $K_{fus}$), we recorded each mitochondria's volume and its next operation (fission, fusion, none) at the next timestamp (time interval = 10s). Mitochondria volumes were binned using the Freedman-Diaconis rule and the rate of fission/fusion was computed for each bin as noted previously. Data was fit to linear equations of the form y = Ax+B.

## Motion analysis

Using mitochondria lineage maps generated from our movies, we calculated MSD from tracks where mitochondria did not undergo either fission or fusion. If a mitochondria underwent fission/fusion, we identified only the trajectories of the mitochondria where the event did not occur. For example, if a mitochondria had no interactions for 10 time steps, underwent fission, and both resulting mitochondria had no interactions for an additional 10 time steps, all three 10 timestep trajectories would be evaluated independently as different objects. Since the center of mass of each mitochondria structure was used as a reference point for location, this scheme was necessary to reduce signal artifact in large changes in mitochondria center of mass due to fusion/fission. Using these isolated trajectories, we computed $MSD = \frac{1}{N} \sum_{i=1}^{N} |x_i(t) - x_i(t=0)|^2$, where $x_i(t=0)$ is the reference position of the $i$-th mitochondria, and $x_i(t)$ is the position of the $i$-th mitochondria at time $t$. The diffusive coefficient was determined by fitting the slope of the MSD curve to $MSD = D\,t$, where D is the diffusive coefficient and $t$ is time.

To determine the diffusive coefficient versus the volume of the mitochondria, mitochondria volume at the start of each isolated mitochondria trajectories was linked to each trajectory. The volume distribution was binned using Freedman-Diaconis, and for each volume bin, MSD and D was computed from individual mitochondria trajectories. Data was fit to linear equations of the form $y = Ax+B$.

To calculate the MSD of DUMPs with respect to their host DUMP$^+$ mitochondria, the trajectory of each DUMP is corrected using the drift of their host mitochondria. If the position of DUMP at time $t$ is $p_{DUMP}(t)$ and the position of the host mitochondria is $p_{mito}(t)$, we compute $MSD = \frac{1}{N} \sum_{i=1}^{N} \|(p_{DUMP,i}(t) - p_{mito}(t)) - (p_{DUMP,i}(t=0) - p_{mito}(t))\|^2$, where $i$ is the $i$-th

DUMP with its associated mitochondria host. Likewise, the diffusive coefficient was determined by fitting the slope of the MSD curve to: $MSD = D\,t$.

We observed that mitochondria primarily display directed motion during specific phases of the cell cycle, particularly when they are transported through the bud neck into the bud (Fig 1G and 1H). Apart from these phases, distinguishing between directed and diffusive motion is difficult due to the seemingly random nature of their movement. To capture the primary behavior of mitochondria along the cell cortex before their transport and passage through the bud neck, we opted to model their movement mainly as diffusive motion within a 2D plane (see methods–"Building the Simulation" and "Inheritance Algorithm").

## Fission/Fusion placement preference analysis

To reference placement of fission/fusion sites with respect to the mitochondria structure, we created 3D skeleton backbones of the mitochondria structure. Skeletonization of mitochondria was achieved via 3D medial axis thinning of each mitochondria's 3D binary segmentation. The same binary segmentation was used to generate a 3D isosurface mesh. This procedure was applied frame by frame to time-lapse images of mitochondria to produce a dynamic 3D model.

To quantitatively map where fission/fusion sites on the mitochondria are, only movies with time-intervals of under 10 seconds were used. Loss of event resolution occurred when longer time intervals were used because multiple fission/fusion events could occur.

Given a mitochondria M that underwent fission, the fission site was determined as the midpoint between the closest point on each of the resulting two mitochondria generated from fission. This fission site position was mapped to the skeletonization of M, S. If M contained any DUMP(s), DUMP position was mapped to segments of S, $s_i$, based on whether $s_i$ passed through DUMP. Using the annotated S, we referenced positioning of fission sites with respect to the DUMP boundary or end of each mitochondrial branch. Normalized fission placement was calculated as distance from the DUMP boundary to the fission site divided by the distance to the end of the mitochondria. Distance was calculated along the skeleton of M. If M contained any branching network structures, the distance was computed up to the first branch point to avoid artificially augmenting bias in normalizing fission site placement (Fig 3B). If fission occurred between two DUMPs, the boundary of a randomly selected DUMP was defined as 0, with the other defined as 1 (Fig 3F).

Given mitochondria $M_1$ and $M_2$ that underwent fusion, mitochondrial fusion sites were identified as the closest points on the surfaces of $M_1$ and $M_2$ prior to fusion. These points were referenced to the mitochondria skeletons of $M_1$ and $M_2$ respectively, with an identical scheme for referencing where DUMP(s) are located. As with fission placement normalization, fusion placement normalization was calculated in a similar manner (S8C, S8F and S8I Fig). When two DUMP$^+$ mitochondria underwent fusion, the larger volume mitochondria was used for determining fusion placement (S8I Fig).

For both fission and fusion in mitoFluc cells, the boundary of DUMP was defined as zero. If fission occurred between two DUMPs, one of the DUMP were randomly selected and its boundary was defined as zero.

For WT mitochondria with no DUMP landmarks, position normalization of fission/fusion placement was calculated by normalizing the end-to-end (or branch point) distance of the mitochondria to 1, along with where fission/fusion occurred (Figs 3D and S8C).

## Puncta mitochondria-region localization analysis

GFP-Dnm1 and GFP-Mdv1 puncta structures were identified using Laplacian of Gaussian filter in 3D images. Each puncta was assigned to the closest mitochondria isosurface within 1μm

and mapped onto a vertex on the mesh. Puncta surface density was determined by dividing the number of puncta per structure by the surface area of the mesh.

For measuring the local concentration of puncta in specific regions of DUMP+ mitochondria, each DUMP binary segmentation was used to generate 3D isosurface mesh. Vertices on the DUMP mesh were then mapped to vertices within 0.5μm on its host mitochondria's mesh. The surface area of the DUMP+ mitochondria was then classified into two surfaces: (1) a surface where vertices were matched to the DUMP projection mesh and (2) no match. Puncta assignment to the DUMP+ mitochondria mesh was similarly classified depending on if their mapped vertex on DUMP+ was mapped to a DUMP projection mesh. The local surface area of these mesh subsets was then used to compute the puncta concentration in specific regions of DUMP+ mitochondria.

This method enables the identification of specific mitochondrial regions and the calculation of puncta per unit surface area within these areas. It provides a more accurate representation of the local concentration of fission machinery, as structures like Dnm1, while not exclusively localized around DUMP, can also be present on mitochondrial structures not immediately adjacent to DUMPs.

## Colocalization analysis

To assess the degree of co-localization between the outer mitochondrial marker (Tom70-GFP) and matrix markers (e.g., MTS-mCherry, MTS-Fluc(WT)-mCherry, and MTS-FlucSM-mCherry), we isolated the GFP and mCherry channels from 3D stack images of individual cells. For each cell, the relationship in intensity between the two channels was evaluated using Pearson's correlation coefficient (PCC), with each PCC value recorded to measure the strength of association between the channels.

## Quantification of GFP-Dnm1 dissociation from mitochondria

To quantify the extent of GFP-Dnm1 dissociation from the mitochondria into cytosolic condensates/puncta, we computed the mean GFP intensity in the cytoplasm and contrasted it with the mean GFP intensity within mitochondrial regions. Both the cytoplasmic and mitochondrial areas were delineated using a 3D mask. The metric for GFP-Dnm1 dissociation was the ratio of the mean GFP intensity in the mitochondria to that in the cytoplasm. A ratio of 1 implies that the Dnm1 concentration on the mitochondria matches its cytosolic concentration. A ratio exceeding 1 signifies a higher Dnm1 concentration on the mitochondria, and a ratio below 1 signifies a higher Dnm1 concentration in the cytoplasm.

## DUMP Mean distance to centroid quantification

DUMP mean distance to centroid was computed for every mitochondria in a mitoFluc expressing cell (constitutive or induced) for only mitochondria with DUMP present. For each mitochondria with $N_{DUMP}$ DUMP(s) each with a position p, the centroid $\bar{p}$ was computed as $\sum_{i=1}^{N_{DUMP}} p_i / N_{DUMP}$. Mean distance to centroid was computed as the average Euclidian norm for all DUMPs as: $\sum_{i=1}^{N_{DUMP}} \|p_i - \bar{p}\|^2 / N_{DUMP}$.

## Probability of DUMP inheritance calculation

To determine the probability of DUMP inheritance, the number of mother-bud cells with DUMP in the buds was divided by the total number of cells sampled, T. To estimate the variability and reliability of this measure, from the dataset of T samples, samples of size T were sampled with replacement 1000 times. The probability of DUMP inheritance was then calculated from each bootstrap sample.

## Building the simulation environment

We conducted our simulation construction and procedures in a manner consistent with existing literature, but tailored parameters to match values derived from our experiments [30]. To model individual mitochondria as networked structures, we needed to estimate their size. Deletion of Fzo1 leads to mitochondrial fragmentation, and we examined the volume distribution of these individual fragmented mitochondria in a Tom70-GFP, Δfzo1 strain (S4A and S4B Fig). From this, we determined the average volume of a single mitochondrion $\bar{V}_{\Delta fzo1}^{mito}$. Each mitochondrion was then represented as a particle with a radius of $r_{particle}$.

To determine the total mitochondrial volume in a budding cell, we measured the average total mitochondrial volume in visibly budding Tom70-GFP cells (S4C Fig). We calculated the number of mitochondrion particles to include in the simulation by dividing the average total mitochondrial volume $\bar{V}_{total}^{mito}$ by $\bar{V}_{\Delta fzo1}^{mito}$, resulting in $N_{total}$ particles. Each of these particles is indivisible but can fuse with each other or separate from a larger network comprised of more particles.

To determine the number of particles that should be passed down in inheritance, we analyzed images of budding cells and measured the average mitochondria volume present in the bud ($\bar{V}_{bud}^{mito}$, S4D Fig). When divided by the volume of a single mitochondrion $\bar{V}_{\Delta fzo1}^{mito}$, we determined $N_{inherit}$ particles were to be inherited per simulation.

To determine the number of particles containing DUMP, we experimentally measured the percentage of mitochondria volume occupied by DUMP in Tom70-GFP cells that constitutively expressed mitoFluc (S4E Fig). Using the average percentage occupancy of mitochondria by DUMP ($P_{occupancy}$, Table 1), we calculated the equivalent volume in our simulation, $N_{DUMP}$.

To determine the surface area of the inner cell cortex surface on which mitochondria traverse, we segmented each budding cell's plasma membrane from DIC images and computed the average surface area of only the mother region, $SA_{mom}$. The simulation box was then set to a box of equivalent area, with a side-length of L. Because the cell inner cortex is a spherical surface, the simulation box was set with periodic boundary conditions on all sides.

## Simulation routine

When the simulation starts, all $N_{Total}$ particles are seeded randomly in the simulation box. To form WT-like random mitochondrial networks, we let the simulation run under WT diffusive properties and fission/fusion probabilities for 30 minutes (simulated time). This time was determined by running the simulation under WT parameters at length (3hrs simulated time) and examining when the simulation converged in number of mitochondria structures (S5A Fig). Each frame of the simulation was the equivalent of 0.1s in real time.

Over the course of the simulation, the occurrence of fission and fusion events was determined based on the probability of fission/fusion conditioned by the volume of the network being examined. These parameters were determined from experimental data and are summarized in Table 1. Two-dimensional particles/networks in the simulation were scaled up to their 3D equivalent volumes and substituted into $K_{fis/fus}^{WT}$, $K_{fis/fus}^{DUMP-}$, or $K_{fis/fus}^{DUMP+}$ depending on if they contained a particle with DUMP.

Fission placement was randomly placed on formed networks unless the biased fission parameter was enabled. When enabled, biased fission occurred between the DUMP marked particles and non-DUMP marked particles. If multiple positions were found, one was selected at random.

For testing the contribution effect of biased fission events over unbiased fission events, the percentage of fission events set as biased was determined by a defined ratio using the values (0,0.2, 0.4, 0.5, 0.6, 0.8, and 1.0). While most mitochondrial fission events near DUMP were biased, a minority did occur randomly (Fig 3C and 3I). Without biased fission (i.e., when the

ratio is 0.0), fission events are distributed randomly throughout the mitochondrial network. Depending on the mitochondrial structure, fission between two DUMPs may occur when the random fission position is selected. Fission occurrence was calculated at each time advancement in the simulation only for mitochondria networks composed of more than one particle.

For testing the effect of adjacent fission events, we tested the scenario where all fission events would occur not between DUMP and non-DUMP marked particles, but rather one more particle out. For example, if three particles "A-B-C" were connected to each other in sequence, and "A" was marked as containing DUMP, adjacent fission would entail a split between particles "B" and "C", resulting in "A-B" and "C", while biased fission would entail a split between particles "A" and "B", resulting in "A", and "B-C". Likewise with biased fission, if multiple positions were found, one was selected at random. Adjacent fission simulations were then compared against simulations where biased fission was enabled or disabled (ratio = 1.0 vs 0.0) (S7 Fig).

Fusion would only occur if two mitochondria structures were within one particle's diameter distance of each other ($2r_{particle}$). When this occurred, probability of fusion was sampled, and if fusion was to occur, placement was determined from experimental distribution of fusion placements (S8C–S8K Fig). To prevent structure overlap, only the two closest particles on the mitochondria undergoing fusion could fuse. Fusion occurs between two mitochondria when they are the closest to each other and if a random value surpassed the experimentally measured threshold for fusion.

The probability of a mitochondrion undergoing fission or fusion at any given timestep was independent of its previous remodeling operations. This means, for instance, that if a mitochondrion underwent fission at one timestep, it was not necessarily more or less likely to undergo fission or fusion in the subsequent timestep. Instead, our model determined these probabilities based on the volume of the mitochondria structure, as derived from our experimental measurements. The model's emphasis is on volume-based probabilities rather than a mitochondria's immediate past fission/fusion event history.

Diffusive motion of mitochondria structures was also calculated dependent on the volume of the network being examined. These parameters are summarized in Table 1. As with fission and fusion, volume was determined by scaling 2D mitochondria network area to their 3D equivalent volume. Scaled volumes were substituted into $D_{WT}$, $D_{DUMP-}$, $D_{DUMP+}$. Displacement was then calculated with a step size of $\sqrt{4\,D\,dt}$. Angle of displacement was randomly chosen. Displacement of the mitochondria structure was applied to the entire mitochondria structure such that all particles in a contiguous network moved in synchrony as a rigid body.

After the initial 30 minute equilibration run, each simulation was run for an additional 90 minutes with modified parameters to conduct a parameter space inquiry. The duration of the additional 90 minute run was determined from the average cell budding cycle time under our experimental conditions at 30˚C [63]. At the 30 minute mark, $N_{DUMP}$ particles were marked, which were seeded as N discrete clusters of particles. The distribution of number of clusters across a set of simulations was determined from experimental measurements in the constitutive mitoFluc strain (S4F Fig). As part of our parameter space inquiry, parameter functions for probability of fission/fusion, diffusive motion, or biased fission could be swapped in a manner such that marked particles could bear DUMP behavioral attributes, continue with WT behavioral attributes, or a mix. Any continuous network that contained a marked particle would also adopt the same behavior—for example, if a marked particle was a part of a larger unmarked-particle structure, the entire structure would exhibit the behavioral properties of the marked particle. A complete list of parameter configurations tested is in Table 2.

To seed N clusters of marked particles in the simulation, an equivalent number of seed points in the mitochondria network were selected. Each seeded particle was marked, and flood

fill was used in marking adjacent particles until the total number of marked particles in the simulation was equal to six. If two initially placed seeds were close enough that additional marked particles would fuse clusters together, a new set of seeds was selected, and the process was repeated until the correct number of clusters was achieved. For evaluating how DUMPs clustered over time, the number of clusters was set equal to $N_{DUMP}$ at t = 1800 in the simulation, thereby creating non-adjacent DUMP clusters of one particle size.

Simulation state was recorded at all timesteps of the simulation where a change in structure was observed. Inheritance and aggregate distribution queries were later run on these recordings.

### Inheritance algorithm

The algorithm for mitochondria inheritance was based on our experimental observations of the three-step mitochondria inheritance process. In a similar way, inheritance in-silico started by identifying the largest connected mitochondria network structure in the simulation (S5B Fig) (blue dashed box). The longest branch of the mitochondria structure was then identified, and from the tip of the branch, $N_{inherit}$ particles were selected using the breadth-first search algorithm. The selected $N_{inherit}$ particles were then considered inherited by the bud. If the inherited structure was smaller than $N_{inherit}$, no additional particles were included to bring the number of inherited particles up to $N_{inherit}$. The inherited structure was later analyzed for DUMP content; if any particle of the inherited particles was marked as containing DUMP, it was considered a failure of aggregate retention or equivalently inheritance of DUMP by the bud cell.

To assess inheritance over the course of the simulation, we applied the inheritance algorithm to every minute after the 30-minute mark. This approach was intended to emulate the potential scenario if mitochondrial inheritance was to occur spontaneously at each respective moment during the simulation. Importantly, no changes to the state of the simulation were necessary to conduct this assessment.

### Model validation

Distribution of mitochondria network volumes *in-silico* were sampled at the end of a 30 minute run under WT parameters, and compared against experimentally measured mitochondria volumes in a Tom70-GFP strain. Volumes for both experimental and simulation data were split into quartiles and ranked against each other to form a QQ plot (S5C Fig), from which residuals were determined (S5D Fig).

To validate *in-silico* versus *in-vivo* event frequency, simulation recordings with only WT parameters were run for the full 120 ($T_{equil}+T_{eval}$) minute run. Since experimental movies were acquired for 5 minutes at 10 second resolution, to ensure a reliable comparison, only the last 5 minutes of each simulation recording was examined. Simulation recordings were also downsampled from their native 0.1 second resolution time interval to 10 seconds to match experimental imaging settings ($dt_{expt}$) [104]. Fission/fusion trajectory maps were then generated from downsampled simulations and the number of fission/fusion events was compared against in-vivo movie data (S5E Fig).

Instructions for running the simulation, along with the code base and visualization scripts, are available in the linked Github repository.

### Quantification and statistical analysis

Please refer to figure captions for sample size descriptions and statistical details. Data is displayed as mean±SEM (standard error of mean) unless otherwise noted. Statistical analysis was performed with R, with p-values below 0.05 considered significant.

## Supporting information

**S1 Fig. Probability of fission/fusion and diffusive motion of mitochondria scales with mitochondria volume and DUMP presence.** (A) Proportion of DUMP+ mitochondria of all mitochondria in mitoFluc cells, n = 413 cells (B) Representative snapshots of DiOC6 stained cells. Top row: WT cells, bottom row: Cells expressing mitoFluc constitutively. (C) Mean DiOC6 fluorescence intensity per mitochondria. n = 206, 263, 141 for WT, DUMP- and DUMP+ mitochondria respectively from 39 WT, 74 mitoFluc cells (WT = mitoBFP only). Mann-Whitney U test, ****$p<0.0001$. (D-F) Bootstrapped probability of fission and fusion per 10s, stratified by mitochondria volume. Shaded ribbons represent 95% of data range (from 0.025 to 0.975 quantile), with line indicating mean. Dashed bold line indicates linear fit. Red = p(Fission) per 10s, blue = p(Fusion) per 10s. (D, G) WT mitochondria = Tom70-GFP only strain. (D) WT mitochondria p(Event) by volume; n = 208 cell movies represented; mitochondria per bin $n{\geq}68$. (E) DUMP- mitochondria p(Event) by volume; n = 134 cell movies; mitochondria per bin $n{\geq}59$. (F) DUMP+ mitochondria p(Event) by volume; n = 134 cell movies; mitochondria per bin $n{\geq}57$. (G-I) Diffusion constant by mitochondria volume. Dashed black line indicates linear fit. (G) Control mitochondria diffusivity by volume; n = 184 cell movies. (H) DUMP- mitochondria diffusivity by volume; n = 106 cell movies. (I) DUMP+ mitochondria diffusivity by volume; n = 119 cell movies. (D-I) Linear fit coefficients are in Table 1.
(TIF)

**S2 Fig. Overexpression of mitoFluc with WT luciferase or MTS-mCherry does not result in DUMP formation or alter mitochondria membrane potential.** (A, C) Scale bar = 5μm shown in merged images. White dashed line demarcates cell boundaries traced from DIC images. (A) Representative images of cells expressing Tom70-GFP (as a mitochondria marker) with MTS-mCherry, mitoFluc (WT), vs. mitoFluc. mitoFluc(WT) contains a non-mutant version of luciferase that is not prone to misfolding. (B) Pearson correlation coefficient of Tom70-GFP signal v.s. mCherry signal for colocalization analysis of mCherry signal with respect to the mitochondria structure. N = 62, 139, 503 for MTS-mCherry, mitoFluc (WT), and mitoFluc (R188Q) respectively. *$p<0.05$, ***$p<0.001$, Wilcoxon rank sum test. (C) Representative snapshots of DiOC6 stained cells. Top row: Cells expressing MTS-mCherry constitutively, bottom row: Cells expressing mitoFluc(WT) constitutively. (D) Mean DiOC6 fluorescence intensity per mitochondria. n = 206, 165, 155 for mitoBFP, MTS-mCherry and mitoFluc(WT) mitochondria respectively from 39 mitoBFP cells, 35 MTS-mCherry cells, and 37 mitoFluc(WT) cells. Mann-Whitney U test, NS>0.9 for all comparisons.
(TIF)

**S3 Fig. Overexpression of MTS-mCherry or mitoFluc with WT luciferase has no effect on mitochondrial fission/fusion rates, while inducible expression of mutant mitoFluc alters these dynamics upon DUMP formation.** Probability of mitochondria to undergo fission (A), fusion (B) per 10s by mitochondria type. Each point represents a single cell's population of the associated mitochondria type. N = 184, 98, 119 single cell movies for Control (Tom70-GFP only), Control + MTS-mCherry, Control + mitoFluc(WT) respectively. Tukey's HSD multiple comparisons test. (A) NS = 0.961 for Control vs. MTS-mCherry, 0.968 Control vs. mitoFluc (WT), and 0.872 MTS-mCherry vs. mitoFluc(WT). (B) NS = 0.998 for Control vs. MTS-mCherry, 0.297 Control vs. mitoFluc(WT), and 0.399 MTS-mCherry vs. mitoFluc(WT). Probability of mitochondria to undergo fission (C), fusion (D) per 10s by mitochondria type over time after induced expression of mitoFluc. $N{\geq}57$ cell movies for all timestamps for induced mitoFluc strain, 121 for Control (EtOH). Black dashed line marks when DUMPs are visible

under microscope in most budding cells. Mean±SEM shown.
(TIF)

**S4 Fig. Additional experimentally derived parameters used for model construction.** (A) Representative snapshot of cell with Tom70-GFP, Δfzo1. Yellow dashed line demarcates cell boundaries traced from DIC images. (B) Mitochondria volume distribution for Δfzo1 mutant. Red vertical line marks mean = $0.12\mu m^3$. n = 141 cells represented. (C-D) Linear regression (blue line) with 95% confidence interval (shaded). Red dashed line indicates mean mother cell volume ($54.0\mu m^3$) used in simulation. n = 346 cells represented. (C) Mother cell volume vs total mitochondria volume in mother and bud. Green dashed line indicates total mitochondria volume ($6.56\mu m^3$) used for simulation. (D) Mother cell volume vs. mitochondria volume in bud. Green dashed line indicates mitochondria volume ($1.83\mu m^3$) used for simulation in inheritance. (E) Distribution of percentage of total mitochondria volume occupied by DUMP. Mean = 11.2% (red-dashed line). (F) Number of DUMPs observed in each cell with constitutive mitoFluc expression, n = 216 cells.
(TIF)

**S5 Fig. Model validation and simulated inheritance algorithm.** (A) Number of discrete mitochondria networks over time. Simulations ran under all WT parameters for motion, probability of fission/fusion, and no targeted fission (n = 100). (B) Two-step algorithm for inheritance implemented in the model. Step (1): largest mitochondria subnetwork by volume is determined (blue dotted box); Step (2): the tip-most N particles (grey dashed ellipse, N = 8 inherited particles for this schematic, for simulation it is set to $N_{inherit}$, Table 1) from the largest branch from the subnetwork is selected as inherited by the bud. Figure is simplified for clarity. (C) Quantile-quantile plot of experimentally measured vs. simulated mitochondria network volume distributions. WT strain (Tom70-GFP) was used as experimental data, simulation data was acquired from running with WT parameters. Volumes are ordered from smallest to largest, with black line marking if both distributions were identical. Blue line represents linear fit to data. N = 1087, 4516 mitochondria for experimental and simulation data, with 346, 1212 cells represented respectively. Slope = 0.977±0.019, $R^2$ = 0.961. (D) Residuals from QQ plot in (C); mean absolute residuals = $0.283 \mu m^3$. (E) Number of fission or fusion events that occurred within a 5-minute window for experimental (n = 184) vs simulation (n = 1212) data. Mann Whitney U-test NS p = 0.25 for fusion, p = 0.9294 for fission.
(TIF)

**S6 Fig. Model predicts biased fission clusters DUMPs.** (A) Representative model simulation illustrating that biased fission facilitates clustering of DUMPs. Specifically, biased fission prevents permanent attachment of DUMPs to non-DUMP regions of DUMP$^+$ mitochondria. Time is shown above each box, with black arrows indicating targeted fission followed by DUMP clustering. Green = DUMP$^+$, red = DUMP$^-$ mitochondria. Particle numbers are for tracking only; grey box indicates periodic boundary condition, with "w" marked on particles that have wrapped around. (B) Number of DUMP(s) in simulation over time with associated (E) bootstrapped probability of DUMP inheritance over time with WT vs mitoFluc parameters (Table 2). (B, E) n = 643, 644 simulations respectively. (C) Number of DUMP(s) in simulation over time with associated (F) bootstrapped probability of DUMP inheritance over time with control ($K_{fis}^{WT}$, $K_{fus}^{WT}$) vs mitoFluc ($K_{fis}^{DUMP+/-}$, $K_{fus}^{DUMP+/-}$) fission/fusion. (C, F) n = 645, 644 simulations respectively, biased fission is on, with $D_{DUMP+/-}$ fixed. (C) Number of DUMP(s) in simulation over time with associated (F) bootstrapped probability of DUMP inheritance over time with control ($D_{WT}$) vs. mitoFluc ($D_{DUMP+/-}$) diffusive motion. (C, F) n = 645, 645 simulations respectively, biased fission is on, with $K_{fis}^{DUMP+/-}$, $K_{fus}^{DUMP+/-}$ fixed. (B-D) Mean±SEM,

(E-G) Mean±SD (standard deviation) are shown.
(TIF)

**S7 Fig. Adjacent Fission also clusters DUMPs.** (A) Number of DUMP(s) in simulation over time with associated (B) bootstrapped probability of DUMP inheritance over time with mitoFluc parameters with different fission placement conditions with mitoFluc parameters (Table 2). Legend for (B) is identical to in (A). Different fission placement conditions are random, adjacent, and biased as described under Methods, "Simulation Routine". (A, B) N = 643, 412, 644 simulations for random, adjacent, and biased fission placement respectively. (A) Mean±SEM, (B) Mean±SD are shown.
(TIF)

**S8 Fig. Location of mitochondria fusion is not altered by DUMP presence.** (A) Representative timeseries showing fission occur between two DUMPs. White arrow indicates site of future fission. (B) Representative timeseries showing tip-to-tip fusion occur between two mitochondria in mitoFluc cells. White arrow indicates site of future fusion. (C, F, I) Future fusion site is marked by black triangle, with dotted projection indicating the length of mitochondria measured. Mitochondria skeleton marked by black solid line spanning green mitochondria, DUMP marked red.(C) Fusion bias normalization procedure for between mitochondria undergoing fusion in WT cells, or between DUMP$^-$ mitochondria in mitoFluc cells. (D) Normalized fusion placement in WT cells. n = 147 cells, 464 fusion events observed. (E) Normalized fusion placement in mitoFluc cells, between DUMP$^-$ and DUMP$^-$ mitochondria. n = 23 cells, 61 fusion events observed. (F, I) Black dotted projection is with respect to the whole mitochondria backbone length (with backbone end closest to DUMP set to 0), while blue dotted projection is with respect to the DUMP boundary (set to 0). (F) Fusion bias normalization procedure for between DUMP$^+$ and DUMP$^-$ mitochondria in mitoFluc cells. Color is matched to (G, H). n = 77 cells, 223 fusion events observed. Normalized fusion placement in mitoFluc cells between DUMP$^+$ and DUMP$^-$ mitochondria normalized to mitochondria backbone (G), and with respect to DUMP location (H). (G, H) Pearson correlation = 0.974 (I) Fusion bias normalization procedure for between DUMP$^+$ and DUMP$^+$ mitochondria in mitoFluc cells. Color is matched to (J, K). n = 15 cells, 23 fusion events observed. Normalized fusion placement in mitoFluc cells between DUMP$^+$ and DUMP$^+$ mitochondria normalized to mitochondria backbone(J), and with respect to DUMP location (K) Normalized fusion placement in mitoFluc cells between DUMP$^+$ and DUMP$^+$ mitochondria. (J, K) Pearson correlation = 0.976.
(TIF)

**S9 Fig. Mdv1 exhibits preferential localization next to DUMP(s), and requires Dnm1 to form puncta around DUMP(s).** Representative GFP-Mdv1 localization in non-mitoFluc (A) vs mitoFluc cells (B). Representative GFP-Mdv1 localization in Δdnm1 cells without (C) and with mitoFluc (D). (E) Mdv1 puncta surface density. Quantification scheme identical to Fig 3A. Control = mitoBFP, GFP-Mdv1 cells; n = 119 cells with 469 mitochondria total. n = 159 cells for mitoFluc, with 583 mitochondria total, 337 which were DUMP$^-$, 246 DUMP$^+$; n = 335 DUMPs were found in DUMP$^+$ mitochondria, which were used to evaluate DUMP projection surface densities. Mann Whitney U test, NS p = 0.948; ***p<0.001; ****p<0.0001. (F) Total number of GFP-Mdv1 puncta per cell with no-mitoFluc vs constitutive mitoFluc expression, n = 119, 159 cells respectively. Mann Whitney U-test, NS p = 0.221. (G) Dnm1 puncta surface density in Δcaf4 cells. Quantification scheme identical to Fig 5E. Control = mitoBFP, GFP-Dnm1, Δcaf4 cells; n = 72 cells with 494 mitochondria total. n = 58 cells with mitoFluc, with 406 mitochondria total, 299 which were DUMP$^-$, 107 DUMP$^+$; n = 142 DUMPs were found in DUMP$^+$ mitochondria, which were used to evaluate DUMP projection surface

densities. Mann Whitney U test, NS p = 0.996; ****p<0.0001. (H) Representative GFP-Dnm1 localization in Δcaf4 cells. (I) Representative GFP-Dnm1 localization in mitoFluc, Δcaf4 cells. All images in the GFP channel are displayed with consistent intensity range, brightness, and contrast settings for accurate comparison.
(TIF)

**S10 Fig. Fis1 and Caf4 localization remains unchanged in the presence of DUMP(s), however, the absence of more than one component of the fission machinery leads to Dnm1 dissociation from the mitochondria.** Representative Caf4-GFP localization in control (A) vs mitoFluc cells (B). Representative Fis1-GFP localization in control (C) vs mitoFluc cells (D). Representative GFP-Dnm1 localization in Δfis1, Δmdv1Δcaf4, Δmdv1Δfis1, without (E) and with (F) mitoFluc. All images in the GFP channel are displayed with consistent intensity range, brightness, and contrast settings for accurate comparison.
(TIF)

**S11 Fig. Deletion of multiple fission machinery components leads to Dnm1 disassociation from mitochondria and its localization in the cytosol.** Ratio of GFP-Dnm1 density on mitochondria over cytoplasm when cells do not have constitutive expression of mitoFluc (A) vs. when mitoFluc is constitutively expressed (B). Horizontal red-dashed line marks a ratio of 1.0, which would indicate that the fluorescence density of GFP-Dnm1 is identical on the mitochondria and cytosol per $\mu m^3$. (A, B) All labels on X axis with "+" indicate additional mutations applied to control strain. (A) Control = mitoBFP, GFP-Dnm1. N = 114, 103, 52, 72, 126, 145 cells for Control, +Δmdv1, +Δcaf4, +Δfis1, +Δmdv1Δfis1, +Δmdv1Δcaf4 respectively. (B) Control = mitoBFP, GFP-Dnm1, mitoFluc. N = 136, 102, 109, 63, 111, 141 cells for Control, +Δmdv1, +Δcaf4, +Δfis1, +Δmdv1Δfis1, +Δmdv1Δcaf4 respectively. Black diamond marks mean.
(TIF)

**S12 Fig. Δdnm1 results in mitochondria lattice-like regions that preferentially sequester DUMP formation.** (A-B) Scale bar = 5μm shown in merged images. White dashed line demarcates cell boundaries traced from DIC images. (A) Representative structured illumination microscopy image of MTSmCherry Δdnm1 cell showing mitochondria net structures, (B) of Tom70-GFP, Δdnm1 with constitutive mitoFluc. (C-F) Bootstrapped probability of observing DUMP in net-like domain of mitochondria. Iterations = 500, sample size = n. (C) Δdnm1 cells with constitutive mitoFluc expression, n = 84 cells. (D) Δdnm1 cells with induced mitoFluc expression, n = 89, 139, 113, 155, 95 for timestamps 60, 90, 120, 150, 180 respectively. Bootstrapped probability of DUMP inheritance in cells expressing mitoFluc constitutively (E) n = 462 cells or induced (F) over time n = 105, 146, 114, 160, 96 for timestamps 60, 90, 120, 150, 180 respectively. Bootstrap sampling from sample population of size n; iterations = 500, sampling size = n. (G) Percentage of total mitochondria volume is lattice-like in Δdnm1 cells. n = 51, 62 cells for control (Δdnm1 only) and Δdnm1 with constitutive mitoFluc, respectively. Two tailed t-test, NS (p = 0.069). (H) Percentage of total mitochondria volume is lattice-like in Δdnm1 cells with inducible mitoFluc after exposure to β-Estradiol. n = 106, 126, 111, 150, 85 for timestamps 60, 90, 120, 150, 180 respectively.
(TIF)

**S13 Fig. Deletion of fission/fusion machinery results in loss of fission/fusion.** (A, B) Probability of fission and fusion per 10 seconds. (A) With no mitoFluc expression. n = 139, 150, 59 cell movies for Δdnm1, Δdnm1Δfzo1, Δdnm1Δmgm1, respectively. (B) With constitutive mitoFluc expression. n = 89, 78, 49 cell movies for Δdnm1, Δdnm1Δfzo1, Δdnm1Δmgm1,

respectively.
(TIF)

## Acknowledgments

We thank Paul Henderson, and Rubab F. Malik for technical assistance, and members of the R.L. and J.L. labs for helpful discussions, Keir C. Neuman for graciously providing access to his lab facilities and microscope while ours was under repair, Abhijit Deb Roy, Alexis Tomaszewski for advice on manuscript preparation.

## Author Contributions

**Conceptualization:** Gordon Sun, Jian Liu, Rong Li.

**Data curation:** Gordon Sun, Christine Hwang, Tony Jung.

**Formal analysis:** Gordon Sun, Christine Hwang, Tony Jung.

**Funding acquisition:** Jian Liu, Rong Li.

**Investigation:** Gordon Sun, Jian Liu, Rong Li.

**Methodology:** Gordon Sun, Jian Liu, Rong Li.

**Project administration:** Gordon Sun, Jian Liu, Rong Li.

**Resources:** Jian Liu, Rong Li.

**Software:** Gordon Sun.

**Supervision:** Jian Liu, Rong Li.

**Validation:** Gordon Sun, Christine Hwang, Tony Jung.

**Visualization:** Gordon Sun.

**Writing – original draft:** Gordon Sun.

**Writing – review & editing:** Gordon Sun, Jian Liu, Rong Li.

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
