## [Decision Letter · Decision Letter 0]

27 Jul 2023

Dear Dr. Liu,

Thank you very much for submitting your manuscript "Biased Placement of Mitochondria Fission Facilitates Asymmetric Inheritance of Protein Aggregates during Yeast Cell Division" for consideration at PLOS Computational Biology.

As with all papers reviewed by the journal, your manuscript was reviewed by members of the editorial board and by several independent reviewers. In light of the reviews (below this email), we would like to invite the resubmission of a significantly-revised version that takes into account the reviewers' comments.

We cannot make any decision about publication until we have seen the revised manuscript and your response to the reviewers' comments. Your revised manuscript is also likely to be sent to reviewers for further evaluation.

Sincerely,

Attila Csikász-Nagy

Academic Editor

PLOS Computational Biology

Jason Haugh

Section Editor

PLOS Computational Biology

Reviewer's Responses to Questions

**Comments to the Authors:**

Reviewer #1: The authors study the asymmetric inheritance of mitochondria between mother and daughter budding yeast cells. Asymmetric inheritance is important to ensure that daughter cells receive healthy mitochondria. The authors combine computational modeling with live cell fluorescent imaging to investigate the mechanism that underlies asymmetric inheritance. In particular, they make use of fluorescently labeled deposits of unfolded mitochondrial proteins (DUMPs) to monitor the retention of misfolded mitochondrial proteins by the mother cell. Their initial experimental investigations revealed that mitochondria containing DUMPs were more likely to undergo fission. Based on this observation, they develop an agent-based model that reproduces asymmetric inheritance. The model predicts the formation of DUMP clusters is important for asymmetric inheritance. Finally, they perform experiments to demonstrate DUMPs form clusters and mitochondria fission is required for efficient DUMP clustering. The data presented in the manuscript seem to provide valuable clues for the mechanism that underlies asymmetric inheritance. However, as discussed below, I am not convinced these data address a key assumption of the agent-based model.

1. Because the paper appears to be more heavily weighted toward experiments than computational studies, I wonder if a different journal (e.g. PLoS Biology or MBoC) might be a more appropriate venue for it.

2. In the section “Biased fission is predicted to underlie the asymmetric retention of DUMP via DUMP clustering” the authors state that in their model “When biased fission was present, fission would only occur between adjacent DUMP marked and non-DUMP particles.” However, their experiments seem to indicate fission also occurs between DUMP sites (Figs. 3F and G). How do the authors reconcile this observation with their model?

3. The assumption that fission only occurs between DUMP and non-DUMP particles appears to be critical for generating DUMP clusters because when two DUMP particles fuse, they are unlikely to dissociate. This assumption also appears to be necessary for the mitochondria containing the DUMP clusters not to be the longest branch, because when a non-DUMP particle associates with the DUMP cluster, it is likely to undergo fission, keeping the size of this branch short. What happens in the simulations if instead of assuming biased fission only occurs between DUMP and non-DUMP particles, fission is simply increased adjacent to DUMP particles? If this is not sufficient to generate DUMP clusters that are retained in the mother, is there a way to show experimentally that two nearby DUMP sites are less likely to undergo fission then equal length portions of the mitochondrial chain that are not between to DUMP sites?

4. Simulation results for DUMP clustering in the form of time series or movies should be presented, so readers can see what the simulation results look like and compare them with the experimental results.

Reviewer #2: Summary:

In this manuscript, the authors seek to define how yeast cells segregate damaged parts of mitochondria to ensure that healthy mitochondria are passed on to progeny cells. To do this, they used imaging to quantify mitochondrial dynamics and a protein aggregation model that is targeted to the mitochondrial matrix to impart damage. Their model predicts that biased mitochondrial fission near the protein aggregates facilitates the clustering of protein aggregates in the mitochondrial matrix, and this process underlies asymmetric mitochondria inheritance. They highlight that impaired mitochondrial fission reduces the rate of aggregate clustering, and they see a similar phenomenon when they delete the inner mitochondrial membrane fusion protein, Mgm1p.

A unique aspect of this work is integrated live-cell experiments and modeling to determine how mitochondrial remodeling underlies asymmetric mitochondria inheritance. They find and clearly demonstrate that mitochondrial dynamics impacts organelle inheritance. I think similar findings have been observed in other organisms, but the combination of yeast genetics and modeling highlights a powerful combination to explore the roles of specific genes in these events. I am generally positive about the findings and the quality of the work. The paper is generally well written and organized. My concerns are highlighted below.

Major concerns:

1. Cells expressed mitochondria-targeted misfolded proteins as a model system. How much overexpression is occurring and can this be controlled? There was no mention or consideration of this when the experiments are introduced or in the methods section.

2. Does the expression of a NON-Mutant fly luciferase distribute uniformly throughout the mitos? And does this not impact mitochondrial dynamics/behavior in the same way? This might distinguish the impact of the gene vs the misfolded protein response.

3. Related to the above point, the DUMP cells (+ and -) have lost membrane potential. This brings in other confounding variables associated with mitophagy potentially. So this may be an amplified fission expt rather than normal rates of fission. And how is the luciferase protein impacting membrane potential? Even when it is supposedly not misfolded?

4. Diffusive motion suggests that mitochondria move randomly in the cell, but they are often moving along microtubules by motors or being pushed by actin. This is not directly incorporated in the diffusion parameter with any vector. But is this achieved by using a 2D plane to limit conformational freedom? And how does this limit the model?

5. It is not clear to me whether the DUMP mitos that preferentially divide/separate are more prone to fuse/interact with adjacent mitochondria in the model. If so, they are essentially stuck in this local cycle of fission and fusion? Is that the idea? And if so, what would be the proposed mechanism in a cellular environment that promotes this cycling?

6. In the data for figure 5, it would be better to have a co-localization measure (Pearson Coefficient or something similar) to assess Dnm1 interactions with DUMP. The number of punctae (N) is less informative since there are many other punctae on the mitochondria. This is done later, but it would be a good practice when showing co-localization in cells.

7. The deletion of Mdv1 and Caf4 are not affecting Dnm1 localization to DUMP sites. The authors show ∆Fis1 in the supplemental data, but there appear to be a number of additional Dnm1 punctae that form at locations other than mitochondria. This complicates the interpretation and the co-localization is less clear in the image presented. Better quantitative assessment of the co-localization would help. Additionally, GFP has been shown to promote assembly in the mammalian homolog of Dnm1p, Drp1. A similar phenomenon might be true in these expts, which is why so many punctae are observed.

8. I don't understand Fig S9. This is supposed to demonstrate the rates of fission and fusion, but I don't see any difference between the averages in the different strains (WT vs deletions). It looks like there are very few events every 10 s. Is there a difference if a longer timepoint is considered? I may just not understand the graph tbh.

9. Perhaps the most interesting results is that you need Mgm1 to get matrix mixing that facilitates the consolidation of the DUMPs. Is the next step to identify the coordination of fission and fusion to isolate these damaged regions? Discussion on the extent to which fusion contributes to this process would be warranted. The discussion focused solely on fission.

Minor comments:

pg 29 - The authors refer to Mdv1 as a "membrane-bound protein", but I believe that this is not true. It associates with the membrane through interactions with Fis1. Mdv1 does not have a membrane binding region to my knowledge.

The authors state that "deletion of CAF4, a gene encoding an Mdv1 paralog [69] that does not form puncta (Fig S7A, B)", but this is actually Fig S6, H-I)

**Have the authors made all data and (if applicable) computational code underlying the findings in their manuscript fully available?**

Reviewer #1: Yes

Reviewer #2: Yes

PLOS authors have the option to publish the peer review history of their article (what does this mean?). If published, this will include your full peer review and any attached files.

Reviewer #1: No

Reviewer #2: **Yes: **Jason Mears
---

## [Decision Letter · Decision Letter 1]

10 Oct 2023

Dear Dr. Liu,

We are pleased to inform you that your manuscript 'Biased Placement of Mitochondria Fission Facilitates Asymmetric Inheritance of Protein Aggregates during Yeast Cell Division' has been provisionally accepted for publication in PLOS Computational Biology.

Best regards,

Attila Csikász-Nagy

Academic Editor

PLOS Computational Biology

Jason Haugh

Section Editor

PLOS Computational Biology

Reviewer's Responses to Questions

**Comments to the Authors:**

Reviewer #2: I have nothing else to add at this time

Reviewer #3: I have read the reviewers comments and revised manuscript and in my opinion, the authors have addressed the concerns of the two previous reviewers.

**Have the authors made all data and (if applicable) computational code underlying the findings in their manuscript fully available?**

Reviewer #2: Yes

Reviewer #3: None

PLOS authors have the option to publish the peer review history of their article (what does this mean?). If published, this will include your full peer review and any attached files.

Reviewer #2: No

Reviewer #3: No

---

## [Editor Report · Acceptance letter]

16 Nov 2023

PCOMPBIOL-D-23-01020R1 

Biased Placement of Mitochondria Fission Facilitates Asymmetric Inheritance of Protein Aggregates during Yeast Cell Division

Dear Dr Liu,

I am pleased to inform you that your manuscript has been formally accepted for publication in PLOS Computational Biology. Your manuscript is now with our production department and you will be notified of the publication date in due course.

With kind regards,

Zsofia Freund
